# Robust Learning via Nested Distributionally Robust Optimization

**Jinyi Huang**[1]   **Jinlong Lei**[1 2]   **Guodong Shi**[3]

## Abstract

Distributionally Robust Optimization (DRO) is widely used to improve model robustness, with existing methods addressing either geometric perturbations (e.g., input shifts) or statistical contamination (e.g., heavy-tailed noise and outliers) effectively. However, these uncertainty sources often co-exist, and coupling them through a single divergence or optimal transport constraint conflates geometric displacement with loss-based outlierness, frequently discarding informative high-leverage samples. We introduce *nested DRO*, a bilevel formulation that combines an outer optimistic $\phi$-divergence cleaning step with an inner pessimistic optimal-transport robustification step, thereby decoupling geometric smoothing from statistical cleaning. We prove that this structure naturally induces a geometry-invariant, loss-based reweighting mechanism that separates outlier suppression from transport-induced regularization. We derive a tractable strong dual for the resulting non-convex problem and show its equivalence to variance-regularized risk minimization, leading to a clear statistical interpretation of the induced weights. Empirical results on synthetic and real datasets demonstrate that nested DRO consistently outperforms geometry-coupled DRO baselines, particularly under heavy-tailed contamination where preserving high-leverage structure is crucial.

## 1. Introduction

Distributionally robust optimization (DRO) has established itself as a rigorous paradigm for training machine learning models that remain reliable under data uncertainty. Existing research typically approaches distributional ambiguity through two distinct lenses. One stream focuses on statistical divergences (Read & Cressie, 2012; Pardo, 2018), which model uncertainty by reweighting probability mass but ignore the underlying geometry of the data space. The other stream relies on geometric metrics, particularly optimal transport (OT) distances (Blanchet et al., 2022; Gao & Kleywegt, 2023), which capture data geometry and provide robustness against feature perturbations. While both approaches have firm theoretical groundings, they are predominantly studied in isolation. Modern machine learning pipelines, however, inevitably face compound uncertainty, where local geometric perturbations (Sinha et al., 2017) co-exist with gross statistical contamination (Huber, 1992).

Relying solely on geometric metrics to handle this compound uncertainty introduces a critical failure mode. Standard optimal transport DRO (OT-DRO) formulations, and even their outlier-robust extensions, typically couple sample reliability directly to transport cost. Under this geometric paradigm, samples that are spatially distant from the distribution center are penalized or discarded as anomalies. This leads to a phenomenon we term geometric conflation, where valid high-leverage points in heavy-tailed or highly anisotropic regimes become geometrically indistinguishable from adversarial outliers. Consequently, geometric-based robustness mechanisms inadvertently suppress valid tail phenomena, sacrificing generalization on hard examples for the sake of stability. Conversely, purely statistical methods lack the geometric awareness to defend against adversarial shifts in feature space. Therefore, decoupling geometric robustification from statistical cleaning within a unified robust learning framework remains a gap in DRO. We aim to fill this gap.

### 1.1. Contributions

We propose $\phi$-divergence constrained optimal transport DRO ($\phi$-OT DRO) as a nested framework that effectively decouples statistical cleaning from geometric smoothing. By constraining the reference measure within a $\phi$-divergence ball, our model introduces a geometry-agnostic cleaning mechanism that identifies outliers based on loss statistics rather than spatial distance. This approach enables the effective distinction between adversarial contaminants and valid

[1]Shanghai Research Institute for Intelligent Autonomous Systems, Tongji University, Shanghai, China [2]Department of Control Science and Engineering, Tongji University, Shanghai, China [3]Faculty of Engineering, The University of Sydney, Sydney, Australia. Correspondence to: Jinlong Lei <leijinlong@tongji.edu.cn>.

*Proceedings of the 43$^{rd}$ International Conference on Machine Learning*, Seoul, South Korea. PMLR 306, 2026. Copyright 2026 by the author(s).

high-leverage points regardless of their geometric position. We further derive a tractable strong dual formulation for this non-convex problem and establish a rigorous theoretical connection to variance regularization, thereby providing both computational efficiency and theoretical justification for the proposed framework. Our specific contributions are threefold:

(i) We propose $\phi$-OT DRO, a nested framework that integrates the distinct paradigms of $\phi$-divergence and optimal transport into a single bilevel optimization. By nesting an optimistic $\phi$-divergence constraint within a pessimistic Wasserstein ambiguity set, our method fundamentally decouples statistical outlier cleaning from geometric smoothing, thereby leveraging the complementary roles of statistical cleaning and geometric robustness.

(ii) We derive a tractable strong dual for this bilevel optimization, revealing a rigorous connection between our robust framework and variance regularization, thereby providing a principled justification for variance-based robustification heuristics.

(iii) We demonstrate that our method significantly outperforms geometry-dependent baselines in geometric conflation regimes, where valid high-leverage points and adversarial outliers are indistinguishable by distance.

As a principled and computationally efficient framework, the nested DRO enables robust learning under compound geometric and statistical uncertainty.

## 1.2. Related Works

**Distributionally Robust Optimization.** Data-driven DRO typically constructs ambiguity sets using either statistical divergences or OT distance. Divergence-based approaches, such as $\chi^2$ and KL divergences, model statistical uncertainty and often admit tractable reformulations for chance-constrained programs (Hu et al., 2013; Jiang & Guan, 2016). These formulations are also rigorously connected to variance regularization (Duchi & Namkoong, 2019; Duchi et al., 2021). Recent works have further focused on the statistical stability of these estimators, proposing algorithms explicitly for bias and variance reduction in DRO training (Mehta et al., 2024).

Conversely, OT-based approaches, particularly Wasserstein DRO, focus on capturing geometric uncertainty (Sinha et al., 2017; Mohajerin Esfahani & Kuhn, 2018; Gao et al., 2024). The resulting objectives naturally induce norm-based regularization, which can be interpreted as robustness against geometric perturbations(Gao et al., 2024). This perspective has also been extended to incorporate structural causal models and individual fairness considerations in Wasserstein DRO (Ehyaei et al., 2024). On the computational front, recent advances include Sinkhorn regularization (Yang et al., 2024) and efficient solvers for non-convex landscapes (Yu et al., 2024; Zhang et al., 2025). However, these paradigms are predominantly studied in isolation, treating them as distinct methodologies rather than integrating them into a nested framework.

**Robustness in DRO.** The DRO formulations discussed above provide foundational tools for uncertainty modeling, and research has been dedicated to applying them to real-world datasets with corruption to address contamination. Existing strategies within the DRO framework generally fall into three streams, depending on the type of contamination they target.

The first stream leverages OT-based constraints to defend against geometric perturbations (Sinha et al., 2017; Gao et al., 2024). However, standard OT formulations can be overly sensitive to gross statistical outliers. The transport cost mechanism compels the model to account for distant samples even when they represent irrelevant noise rather than informative hard negatives.

The second stream employs loss-based filtering or reweighting to handle non-geometric contamination. Approaches such as DORO (Zhai et al., 2021) and SRGD (Kumar et al., 2023) identify and remove a fraction of high-risk data points based on purely statistical criteria. While effective against label noise, they ignore local manifold structure, leaving models vulnerable to adversarial feature perturbations.

The third stream attempts to address the compound uncertainty where geometric shifts and statistical outliers co-exist. Recent works have sought to handle this mixed uncertainty by coupling geometric and statistical constraints. Prominent methods include OR-WDRO (Nietert et al., 2023), which combines total variation with Wasserstein distance, and UOT-DRO (Wang et al., 2024), which utilizes unbalanced optimal transport. Very recently, (Li et al., 2025) proposed a unified framework that integrates robust statistics with Wasserstein DRO, estimating the DRO objective from corrupted data by utilizing robust mean estimation oracles to filter out outliers. Crucially, these methods couple outlier detection directly to feature geometry, implicitly assuming that outliers are always geometrically isolated. In heavy-tailed regimes, this leads to geometric conflation, where valid high-leverage samples (which are geometrically distant but statistically consistent) are erroneously discarded.

From a robust-statistical perspective, our outer $\phi$-divergence step is conceptually aligned with contamination handling through trimming or reweighting (Zhai et al., 2021; Kumar et al., 2024): a small portion of probability mass may be discounted away from statistically inconsistent samples without relying on their geometric location. In particular,

the total-variation instantiation is closely related to classical contamination neighborhoods in robust statistics, most notably Huber's $\epsilon$-contamination model. Viewed through this lens, the role of the outer layer is not to introduce a new cleaning principle in itself, but to provide a geometry-agnostic contamination-handling mechanism that can be composed with inner Wasserstein robustification. Our contribution therefore lies in this nested composition: Nested DRO combines such geometry-agnostic statistical cleaning with an inner Wasserstein robustification step, using OT distance for local geometric smoothing while reserving the outer loss-based $\phi$-divergence step for global contamination-aware cleaning.

## 2. Preliminaries

In this section, we review the mathematical foundations used in our framework, including distributionally robust optimization based on optimal transport and the calculus of $\phi$-divergences. We denote the nominal distribution by $\mathbb{P}_0$ and the loss function by $f_\theta(z)$.

### 2.1. DRO via Optimal Transport

Optimal transport models geometric uncertainty by allowing probability mass to shift locally in the feature space. Given a lower semi-continuous ground cost function $c(z, z')$, the optimal transport distance $W_c(\mathbb{P}, \mathbb{Q})$ represents the minimal cost to transport $\mathbb{P}$ to $\mathbb{Q}$ and is defined as

$$W_c(\mathbb{P}, \mathbb{Q}) = \inf_{\pi \in \Pi(\mathbb{P}, \mathbb{Q})} \mathbb{E}_{(z,z') \sim \pi}[c(z, z')], \qquad (1)$$

where $\Pi(\mathbb{P}, \mathbb{Q})$ is the set of couplings with marginals $\mathbb{P}$ and $\mathbb{Q}$. A standard optimal transport robust formulation minimizes the worst-case expected loss over a constraint

$$\sup_{\mathbb{P}: W_c(\mathbb{P}, \mathbb{P}_0) \leq \rho} \mathbb{E}_{z \sim \mathbb{P}}[f_\theta(z)]. \qquad (2)$$

This formulation effectively smooths the loss landscape with respect to the geometry induced by the cost function and provides robustness against feature perturbations.

### 2.2. $\phi$-Divergences and Conjugate Duality

To quantify statistical discrepancy without geometric bias, we employ $\phi$-divergences. Let $\phi : \mathbb{R} \to \mathbb{R} \cup \{+\infty\}$ be a convex lower semi-continuous function with $\phi(1) = 0$. The $\phi$-divergence between two probability measures $\mathbb{Q}$ and $\mathbb{P}$ is defined as

$$D_\phi(\mathbb{Q} \| \mathbb{P}) \triangleq \int \phi\left(\frac{d\mathbb{Q}}{d\mathbb{P}}(z)\right) d\mathbb{P}(z).$$

To ensure well-definedness when supports differ, we adopt standard conventions where $0\phi(0/0) \triangleq 0$ and $0\phi(a/0) \triangleq a \lim_{t \to \infty}(\phi(t)/t)$ for $a > 0$.

*Table 1.* Common $\phi$-divergences and their Fenchel conjugates $\phi^*(s)$.

| NAME | $\phi(t)$ | $\phi^*(s)$ |
|---|---|---|
| KULLBACK-LEIBLER | $t \log t - t + 1$ | $e^s - 1$ |
| $\chi^2$-DIVERGENCE | $\frac{1}{2}(t-1)^2$ | $\begin{cases} s + \frac{1}{2}s^2, & s > -1 \\ -1/2, & s \leq -1 \end{cases}$ |
| J-DIVERGENCE | $(t-1)\log t$ | NO CLOSED FORM |
| TOTAL VARIATION | $\frac{1}{2}\|t-1\|$ | $\begin{cases} s, & \|s\| \leq \frac{1}{2} \\ -\frac{1}{2}, & s \leq -\frac{1}{2} \end{cases}$ |
| BURG ENTROPY | $-\log t + t - 1$ | $-\log(1-s), \ s < 1$ |

A crucial tool for deriving tractable reformulations involving $\phi$-divergences is the Fenchel conjugate which is defined as

$$\phi^*(s) \triangleq \sup_{t \geq 0}\{st - \phi(t)\}, \qquad s \in \mathbb{R}.$$

Our analysis relies on the scaling property of conjugates where $(\lambda\phi)^*(a) = \lambda\phi^*(a/\lambda)$ for any $\lambda > 0$. For the limiting case $\lambda = 0$, we use the recession function given by the lower semi-continuous extension

$$(0\phi)^*(a) = 0\,\phi^*\left(\frac{a}{0}\right) \triangleq \begin{cases} 0, & a \leq 0, \\ +\infty, & a > 0. \end{cases} \qquad (3)$$

Table 1 lists some common $\phi$-divergences and their conjugates.

## 3. $\phi$-Divergence Constrained OT-DRO

In this section, we introduce the nested DRO framework, designed to effectively handle compound uncertainty by decoupling geometric smoothing from statistical cleaning. We derive a computationally tractable dual formulation for this bi-level problem and analyze its induced robust reweighting mechanism. Finally, we connect the framework to variance regularization and present a practical optimization algorithm.

### 3.1. The Nested DRO Framework

**The Objective.** We consider a supervised learning setting where data samples $z = (x, y)$ are drawn from a joint distribution supported on $\mathcal{Z} = \mathcal{X} \times \mathcal{Y}$, with feature space $\mathcal{X} \subseteq \mathbb{R}^d$ and label space $\mathcal{Y}$. The goal is to learn a predictive model parameterized by $\theta \in \Theta$ that minimizes the loss $f_\theta(z) \triangleq \ell(h_\theta(x), y)$. In this work, we specifically focus on a robust learning setting where local geometric perturbations co-exist with statistical contamination. That is, the data suffers from the following two problems simultaneously: individual inputs may move slightly due to noise or small attacks, and the dataset also contains statistical anomalies.

To address this compound uncertainty, we propose the following nested bilevel structure, which combines an inner geometric robustification step with an outer statistical cleaning step in the following min-min-max nested DRO problem:

$$
\begin{aligned}
&V_\phi(\mathbb{P}_0; \rho, \epsilon) \triangleq \\
&\inf_{\theta \in \Theta} \underbrace{\inf_{\mathbb{P}_D : \mathcal{D}_\phi(\mathbb{P}_D \| \mathbb{P}_0) \leq \epsilon}}_{\text{Statistical Cleaning}} \underbrace{\sup_{\mathbb{P}_W : W_c(\mathbb{P}_W, \mathbb{P}_D) \leq \rho}}_{\text{Geometric Smoothing}} \mathbb{E}_{z \sim \mathbb{P}_W}[f_\theta(z)].
\end{aligned} \quad (4)
$$

This formulation departs from conventional approaches by explicitly structuring the ambiguity set into two distinct layers. Computationally, for a fixed intermediate distribution $P_D$, the framework first evaluates the inner worst-case geometric perturbation and then optimizes the outer contamination-aware cleaning step over $P_D$. Conceptually, the outer $\phi$-divergence layer determines the cleaned anchor distribution around which the inner Wasserstein robustification acts.

**The Inner Geometric Smoothing Layer.** The inner maximization layer handles geometric uncertainty. For a fixed intermediate distribution $\mathbb{P}_D$, the operator $\sup_{\mathbb{P}_W}$ identifies the worst-case distribution subject to an OT cost constraint relative to $\mathbb{P}_D$. This step ensures robustness against local feature perturbations and effectively smooths the loss landscape with respect to the underlying geometry. To ensure that transport occurs only within the feature space without altering semantic labels, we adopt a structured ground cost function $c : \mathcal{Z} \times \mathcal{Z} \to \mathbb{R}_+$, with $\mathcal{Z} = \mathcal{X} \times \mathcal{Y}$, defined as

$$
c(z, z') = c_\mathcal{X}(x, x') + \infty \cdot \mathbf{1}\{y \neq y'\}. \quad (5)
$$

The term $c_\mathcal{X} : \mathcal{X} \times \mathcal{X} \to \mathbb{R}_+$ captures the geometry of the feature space, while the infinite penalty prohibits transport across different labels.

**The Outer Statistical Cleaning Layer.** The outer minimization layer should be interpreted as a contamination-cleaning step rather than as an additional worst-case robustness layer. The operator $\inf_{\mathbb{P}_D}$ searches for an optimal intermediate distribution within a divergence ball of radius $\epsilon$ centered at the nominal data $\mathbb{P}_0$. In contrast to the pessimistic inner layer, this outer layer is optimistic. It enables the framework to implicitly filter out statistically anomalous samples by reweighting the distribution based on loss consistency.

**Structural Decoupling.** By nesting the pessimistic OT cost constraint inside the optimistic divergence constraint, our framework achieves a fundamental structural decoupling of uncertainty. The inner layer provides geometric robustness for valid high-leverage points, while the outer layer acts as a geometry-agnostic filter for outliers. This separation allows the model to distinguish between informative hard examples and harmful contaminants based on their statistical consistency rather than their spatial location.

The order of the two layers in (4) is essential. By placing the optimistic $\phi$-divergence minimization outside the Wasserstein maximization, the model first selects a statistically cleaned intermediate distribution $P_D$ relative to the nominal distribution $P_0$, and only then applies geometric worst-case perturbations around that cleaned anchor. This prevents the transport-based robustness layer from conflating geometric distance with statistical abnormality. By contrast, if the order were reversed so that the Wasserstein adversary acts first, then geometric perturbations would be handled before statistical cleaning, and high-leverage but valid samples could still be entangled with true contaminants. Any subsequent $\phi$-based cleaning would then occur only after this coupling has already taken place, yielding at best a cleaned adversarial distribution rather than the structural decoupling pursued here.

### 3.2. Addressing Computational Tractability

Direct optimization of the nested DRO formulation in (4) is computationally intractable, as it involves searching over an infinite-dimensional space of probability measures. To enable practical implementation, the following result transforms this measure-theoretic problem into a finite-dimensional dual formulation that admits stochastic gradient descent, with the inner maximization handled via numerical approximation.

**Theorem 3.1** (Computationally Tractable Dual Formulation)**.** *Let $f_\theta : \mathcal{Z} \to \mathbb{R}_+$ and $c : \mathcal{Z} \times \mathcal{Z} \to \mathbb{R}_+$ be continuous for all $\theta \in \Theta$. Assume that $\mathcal{Z}$ is compact and that $c(z', z) = 0$ if and only if $z' = z$. Then for any distribution $\mathbb{P}_0$ and any $\rho, \epsilon > 0$, the objective $V_\phi(\mathbb{P}_0; \rho, \epsilon)$ as defined in (4) is equivalent to:*

$$
\begin{aligned}
\inf_{\theta \in \Theta, \lambda \geq 0} \sup_{(\eta, \mu) \in \Lambda_{\phi, \psi_{\theta, \lambda}}} & \left\{ \lambda\rho + \mu - \eta\epsilon \right. \\
& \left. - \eta\, \mathbb{E}_{\mathbb{P}_0}\left[ \phi^*\left( \frac{\mu - \psi_{\theta, \lambda}(z)}{\eta} \right) \right] \right\},
\end{aligned} \quad (6)
$$

*where $\phi^*$ denotes the Fenchel conjugate of $\phi$, and $\psi_{\theta, \lambda}(z)$ is the robust loss function defined by*

$$
\psi_{\theta, \lambda}(z) \triangleq \sup_{z' \in \mathcal{Z}} \left\{ f_\theta(z') - \lambda\, c(z', z) \right\}, \quad (7)
$$

*while the feasible set $\Lambda_{\phi, \psi_{\theta, \lambda}}$ is given by*

$$
\begin{aligned}
\Lambda_{\phi, \psi_{\theta, \lambda}} \triangleq \Big\{ &(\eta, \mu) : \eta \geq 0, \\
&\mu - \psi_{\theta, \lambda}(z) - \eta \lim_{t \to \infty} \frac{\phi(t)}{t} \leq 0, \ \forall z \in \mathcal{Z} \Big\}.
\end{aligned} \quad (8)
$$

We provide the detailed proof in Appendix A. Theorem 3.1 establishes that the worst-case risk is equivalent to a dual objective defined over the low-dimensional scalars $(\lambda, \eta, \mu)$.

In this formulation, the dual variable $\lambda \geq 0$ corresponds to the OT cost constraint $W_c(\mathbb{P}_W, \mathbb{P}_D) \leq \rho$, while $\eta$ and $\mu$ act as the Lagrange multipliers associated with the $\phi$-divergence constraint $\mathcal{D}_\phi(\mathbb{P}_D \| \mathbb{P}_0) \leq \epsilon$ and the probability normalization constraint, respectively.

Although the feasible set (8) may appear complex, it simplifies to the standard constraints $\eta \geq 0$ and $\mu \in \mathbb{R}$ for common divergences such as KL and $\chi^2$, whose conjugate functions have full domains. The general form involving the limit of $\phi(t)/t$ is explicitly maintained to safeguard against domain violations for divergences with bounded conjugate domains, such as the Burg entropy, thereby ensuring that the dual objective remains finite.

Moreover, evaluating the robust surrogate $\psi_{\theta,\lambda}(z)$ is feasible under standard regularity assumptions. When the feature space is bounded and both the loss and transport cost are continuous and differentiable with respect to the input, the inner maximization defines a continuous optimization over the bounded feature space. Although finding a global maximum is challenging for non-convex models, it can be effectively approximated using iterative first-order methods such as projected gradient descent. Crucially, the dual objective (6) depends on the data solely through expectations with respect to the nominal distribution, which permits the use of standard stochastic gradient estimation. Consequently, Theorem 3.1 transforms the computationally intractable worst-case search over the infinite-dimensional space of probability measures into a tractable finite-dimensional optimization problem.

### 3.3. Emergence of Robust Reweighting

The nested DRO framework is constructed by simultaneously employing two distinct ambiguity sets: the inner OT-based ambiguity set models geometric perturbations adversarially, while the outer divergence set serves as an optimistic contamination-cleaning mechanism. Regarding the optimality conditions of the statistical cleaning subproblem

$$\inf_{\mathbb{P}_D : \mathcal{D}_\phi(\mathbb{P}_D \| \mathbb{P}_0) \leq \epsilon} \mathbb{E}_{\mathbb{P}_D}[\psi_{\theta,\lambda}(z)] \tag{9}$$

which corresponds to the outer layer of our nested DRO framework as analyzed in Appendix A, the following result reveals how an intrinsic reweighting mechanism emerges naturally from the framework.

**Theorem 3.2** (Optimal Reweighting Schemes)**.** *Let $\psi_{\theta,\lambda}(z)$ be the robust surrogate loss defined in* (7)*. Suppose that Slater's condition holds for subproblem* (9)*, and the uncertainty radius $\epsilon$ is sufficiently small to ensure that the optimal Lagrange multiplier satisfies $\eta^* > 0$. Then the optimal intermediate distribution $\mathbb{P}_D^*$ admits the following closed form representations.*

 *(i) For the $\chi^2$-divergence where $\phi(t) = \frac{1}{2}(t-1)^2$, the*

*optimal distribution follows a linear reweighting*

$$d\mathbb{P}_D^*(z) = \left(1 - \frac{\psi_{\theta,\lambda}(z) - \mathbb{E}_{\mathbb{P}_0}[\psi_{\theta,\lambda}(z)]}{\eta^*}\right)_+ d\mathbb{P}_0(z).$$

*(ii) For the KL-Divergence where $\phi(t) = t \log t - t + 1$, the optimal distribution follows an exponential reweighting*

$$d\mathbb{P}_D^*(z) = \frac{e^{-\psi_{\theta,\lambda}(z)/\eta^*}}{\mathbb{E}_{\mathbb{P}_0}\left[e^{-\psi_{\theta,\lambda}(z)/\eta^*}\right]} d\mathbb{P}_0(z).$$

We provide the detailed proof and the complete set of regularity conditions in Appendix B.

**Mechanism via Nested Ambiguity Structure.** Theorem 3.2 reveals the intrinsic behavior of our nested model, which directly induces a reweighting mechanism. Specifically, the inner maximization handles local feature perturbations to generate the robust surrogate $\psi_{\theta,\lambda}(z)$ defined in (7). This surrogate represents the original loss inflated by the local sensitivity of the sample. It essentially measures the worst case loss within a small geometric neighborhood. Subsequently, the outer divergence minimization operates on these surrogates to filter out statistical noise. This interpretation also covers total variation distance, for which the optimistic outer step can be viewed as a geometry-agnostic trimmed or reweighted cleaning rule on $\psi_{\theta,\lambda}$. This two step process automatically assigns lower probability mass to samples where this robustified loss remains high. Consequently, the model naturally identifies and suppresses data points that are inconsistent with the learned distribution.

**Decoupling Statistical Validity from Geometry.** Crucially, this reweighting relies solely on the robust loss value and remains independent of the sample location in the feature space. A valid high leverage point may be geometrically distant but statistically consistent with the model. It retains high weight because its robust loss remains low. Conversely, an adversarial contaminant characterized by statistical inconsistency is effectively suppressed. This confirms that our framework successfully decouples statistical validity from geometric position.

**The Role of Dual Variable $\eta^*$.** The scalar $\eta^*$ serves as the adaptive sensitivity parameter governed by the uncertainty radius $\epsilon$. From the perspective of convex duality, $\eta^*$ is the Lagrange multiplier for the constraint on the divergence between the distributions. An inverse relationship exists where a larger radius implies a higher suspected contamination and necessitates a smaller $\eta^*$. In the reweighting formulas, a smaller $\eta^*$ amplifies the downweighting term and forces the probability mass to concentrate more sharply on low loss samples. This mechanism allows the model to adaptively regulate the aggressiveness of the cleaning process by adjusting the uncertainty radius $\epsilon$ based on the presumed noise level.

## 3.4. Variance Regularization and Algorithm

Building on the general dual framework established in Theorem 3.1, we now instantiate the divergence function $\phi$ to derive a tractable objective suitable for numerical optimization. Specifically, we focus on the $\chi^2$-divergence, defined by $\phi(t) = \frac{1}{2}(t-1)^2$. Unlike the general case, this quadratic form allows us to solve the inner maximization problem analytically, yielding an exact mean-variance regularized objective. We formally state this closed-form result in the following theorem.

**Theorem 3.3** ($\chi^2$-Divergence Dual). *Let $V_{\chi^2}(\mathbb{P}_0; \rho, \epsilon)$ denote the objective $V_\phi(\mathbb{P}_0; \rho, \epsilon)$ in (4) with $\phi$ specified as the $\chi^2$-divergence, i.e., $\phi(t) = \frac{1}{2}(t-1)^2$. Then, under the conditions of Theorem 3.1, the objective admits the following equivalent form:*

$$V_{\chi^2}(\mathbb{P}_0; \rho, \epsilon) = \inf_{\theta \in \Theta, \lambda \geq 0} \Big\{ \lambda\rho + \mathbb{E}_{\mathbb{P}_0}[\psi_{\theta,\lambda}] - \sqrt{2\epsilon \operatorname{Var}_{\mathbb{P}_0}(\psi_{\theta,\lambda})} \Big\}, \tag{10}$$

*where $\psi_{\theta,\lambda}(z)$ is defined in (7).*

We provide the detailed proof in Appendix C. The objective function (10) establishes a direct theoretical link between nested DRO and variance regularization. This formulation allows for a clear statistical interpretation of the induced robustness. This reformulation is also in the spirit of Rockafellar-style convex-analytic relaxations as it recasts the original nested distributional optimization problem into a tractable dual representation through auxiliary dual variables and conjugate arguments.

**Variance-Regularized Outlier Suppression.** In contrast to standard distributionally robust adversarial training (Sinha et al., 2017), our derived formulation (10) explicitly subtracts a variance term. This distinction arises because our outer layer performs an optimistic reweighting ($\inf_{\mathbb{P}_D}$) for statistical cleaning. Consequently, the term $-\sqrt{2\epsilon \operatorname{Var}_{\mathbb{P}_0}(\psi_{\theta,\lambda})}$ represents a variance-driven cleaning gain, quantifying the expected reduction in loss achieved by adaptively downweighting high-loss outliers within the $\chi^2$-divergence radius $\epsilon$. Unlike the additive penalties in standard robust learning, this negative term reflects the model's adaptive cleaning capability. The coefficient $\sqrt{2\epsilon}$ serves as a cleaning sensitivity parameter that governs the intensity of outlier rejection. A larger $\epsilon$ expands the uncertainty budget for statistical screening, enabling the optimization to more effectively discount high-loss outliers to achieve a refined risk across the distribution support. In the limit as $\epsilon \to 0$, this variance-driven cleaning gain vanishes, and the method degenerates to the standard DRO dual formulation.

**Regime of Validity.** The radius $\epsilon$ acts as a sensitivity coefficient controlling the trade-off between performance (the

---

**Algorithm 1** Variance-Regularized Nested DRO

---

1: **Input:** Dataset $\mathcal{D}$, model $f_\theta$, cost function $c$, radii $\rho, \epsilon$, step sizes $\eta_\theta, \eta_\lambda, \eta_{\text{pgd}}$, PGD steps $K$, batch size $B$.
2: **Initialize:** Model parameters $\theta$, dual variable $\lambda \geq 0$.
3: **for** $t = 1$ **to** $T$ **do**
4: $\quad$ Sample mini-batch $\mathcal{B} = \{(x_i, y_i)\}_{i=1}^B$ from $\mathcal{D}$
5: $\quad$ Initialize feature perturbations $x_i^{(0)} \leftarrow x_i$ for all $i \in \{1, \dots, B\}$
6: $\quad$ **for** $k = 1$ **to** $K$ **do**
7: $\quad\quad$ $g_i \leftarrow \nabla_x \big( f_\theta(x_i^{(k-1)}, y_i) - \lambda c(x_i^{(k-1)}, x_i) \big)$
8: $\quad\quad$ $x_i^{(k)} \leftarrow \Pi_{\mathcal{X}} \big( x_i^{(k-1)} + \eta_{\text{pgd}} \cdot g_i \big)$
9: $\quad$ **end for**
10: $\quad$ Set worst-case $z_i^* \leftarrow (x_i^{(K)}, y_i)$
11: $\quad$ $\psi_i \leftarrow f_\theta(z_i^*) - \lambda c(x_i^{(K)}, x_i)$
12: $\quad$ $\bar{\psi} \leftarrow \frac{1}{B} \sum_{i=1}^B \psi_i, \quad \operatorname{Var}_\psi \leftarrow \frac{1}{B-1} \sum_{i=1}^B (\psi_i - \bar{\psi})^2$
13: $\quad$ Compute Objective: $J \leftarrow \lambda\rho + \bar{\psi} - \sqrt{2\epsilon \cdot \operatorname{Var}_\psi}$
14: $\quad$ Update parameters: $\theta \leftarrow \theta - \eta_\theta \nabla_\theta J$
15: $\quad$ Update dual variable: $\lambda \leftarrow \max(0, \lambda - \eta_\lambda \nabla_\lambda J)$
16: **end for**
17: **Output:** Robust model parameters $\theta^\star$

---

mean term) and stability (the variance term). While Theorem 3.3 provides a clear statistical interpretation, this closed-form representation is strictly valid in the small uncertainty regime. Practically, setting a small $\epsilon$ is necessary to ensure effective training and prevent numerical divergence.

**Algorithm.** Based on the variance-regularized representation derived in Theorem 3.3, we propose a practical optimization procedure detailed in Algorithm 1.

The training iteration proceeds in two distinct phases. First, to address geometric uncertainty, we estimate the robust surrogate $\psi_{\theta,\lambda}(z)$ by solving the inner maximization via multi-step Projected Gradient Descent (PGD). This step captures local worst-case perturbations. Second, distinct from standard adversarial training, which minimizes the simple expected value of these surrogates, our method explicitly aggregates batch statistics to compute both the empirical mean and variance. We then construct the mean-variance objective $\lambda\rho + \mathbb{E}_{\mathbb{P}_0}[\psi] - \sqrt{2\epsilon \operatorname{Var}_{\mathbb{P}_0}(\psi)}$ directly. Finally, we minimize this objective using Stochastic Gradient Descent (SGD). Through these gradient updates, the model automatically suppresses the influence of high-variance outliers, thereby enforcing the stability-performance trade-off dictated by the radius $\epsilon$.

Although Theorem 3.3 is defined at the population level, SGD uses mini-batch variance estimates, inducing stochastic noise akin to gradient noise. Our experimental evaluations in Section 5 further demonstrate that in regimes where high loss volatility might induce numerical instability through overly aggressive variance-driven cleaning, the

adaptive radius strategy (Appendix F) effectively preserves optimization stability across diverse experimental settings.

*Remark* 3.4 (Universality of Variance Regularization). It is worth noting that the variance regularization mechanism derived in Theorem 3.3 is not restricted to the quadratic geometry of the $\chi^2$-divergence. As formalized in Corollary D.3, in the small uncertainty regime where $\epsilon \to 0$, the KL-divergence objective $V_{\mathrm{KL}}$ admits an asymptotically identical variance-regularized expansion:

$$V_{\mathrm{KL}} = \inf_{\theta, \lambda \geq 0} \left\{ \lambda \rho + \mathbb{E}_{\mathbb{P}_0}[\psi_{\theta, \lambda}] \right. $$
$$\left. - \sqrt{2\epsilon \, \mathrm{Var}_{\mathbb{P}_0}(\psi_{\theta, \lambda})} + \mathcal{O}(\epsilon) \right\}.$$

This result demonstrates that the variance-driven cleaning effect is not an artifact of the $\chi^2$-divergence, but instead reflects a consistent operational principle that extends to other major divergence families within the nested DRO framework.

## 4. Generalization Certificate

While Theorem 3.1 and Theorem 3.3 establish the computational tractability of the nested DRO problem, they are formulated on the true distribution $\mathbb{P}_0$. In practice, we only have access to an empirical distribution $\hat{\mathbb{P}}_n$ constructed from $n$ i.i.d. samples. Therefore, it is crucial to certify that the solution obtained from the empirical data generalizes well to the unseen population. To this end, we study the excess risk defined as the difference between the population objective $V_\phi(\mathbb{P}_0; \rho, \epsilon)$ and its empirical counterpart $V_\phi(\hat{\mathbb{P}}_n; \rho, \epsilon)$.

Our analysis relies on the strong duality established in (6), which allows us to shift the complexity analysis from the intractable infinite-dimensional primal ambiguity sets to the finite-dimensional dual objective. To ensure the validity of the generalization bound, we introduce standard regularity assumptions regarding the loss function and the divergence conjugate.

**Assumption 4.1.** The original loss function is uniformly bounded, i.e., $0 \leq f_\theta(z) \leq M$ for all $\theta \in \Theta, z \in \mathcal{Z}$. Furthermore, to ensure the compactness of the dual space, we assume the robust loss $\psi_{\theta, \lambda}(z)$ has a uniformly non-degenerate variance, i.e., there exists $\sigma^2 > 0$ such that $\mathrm{Var}_{\mathbb{P}_0}(\psi_{\theta, \lambda}(z)) \geq \sigma^2$ for all $\theta \in \Theta$ and $\lambda \in [0, M/\rho]$.

**Assumption 4.2.** The Fenchel conjugate $\phi^*$ is locally Lipschitz continuous on its effective domain. That is, for any compact subset $\mathcal{K} \subset \mathrm{dom}(\phi^*)$, there exists a constant $L_{\mathcal{K}} > 0$ such that $\phi^*$ is $L_{\mathcal{K}}$-Lipschitz on $\mathcal{K}$.

**Assumption 4.3.** There exist constants $t_0 \geq 1, c_1 > 0, c_2 \geq 0$, and $q > 0$ such that $\phi^*(t) \geq c_1 t^{1+q} - c_2$ for all $t > t_0$.

These regularity conditions are standard in statistical learn-

ing theory to ensure the concentration of measure. Crucially, Assumption 4.3 guarantees the coercivity of the dual objective, thereby ensuring that the search space for the optimal dual variables remains compact, which is a necessary condition for establishing uniform convergence via Rademacher complexity. In the context of our nested DRO framework, these assumptions are not restrictive; they are naturally satisfied by the primary divergences employed. Specifically, the conjugate of the $\chi^2$-divergence exhibits quadratic growth, while that of the KL-divergence exhibits exponential growth. Furthermore, both conjugates are smooth and satisfy the local Lipschitz continuity required by Assumption 4.2 on any compact domain, ensuring that the theoretical bounds hold in practical implementation.

Building on these regularities, we establish a uniform convergence guarantee based on the Rademacher complexity of the robust loss class.

**Theorem 4.4.** *Let $f_\theta : \mathcal{Z} \to \mathbb{R}_+$ and $c : \mathcal{Z} \times \mathcal{Z} \to \mathbb{R}_+$ be continuous for all $\theta \in \Theta$. Assume that the domain $\mathcal{Z}$ is compact, the parameter space $\Theta$ is compact, and $c(z', z) = 0$ if and only if $z' = z$. Let $\Psi_{\Theta, \Lambda} \triangleq \{z \mapsto \psi_{\theta, \lambda}(z) \mid \theta \in \Theta, \lambda \geq 0\}$ be the class of robust loss functions defined in (7). Then under Assumptions 4.1-4.3, for any $\delta \in (0, 1)$ and $\epsilon, \rho > 0$, with probability at least $1 - \delta$ over the draw from $\mathbb{P}_0$ of $n$ i.i.d. samples, the following bound holds:*

$$\left| V_\phi(\mathbb{P}_0; \rho, \epsilon) - V_\phi(\hat{\mathbb{P}}_n; \rho, \epsilon) \right| \leq$$
$$2L_{\phi^*} \mathfrak{R}_n(\Psi_{\Theta, \Lambda}) + \mathcal{O}\left( \sqrt{\frac{\log(1/\delta)}{n}} \right).$$

*where $\mathfrak{R}_n(\Psi_{\Theta, \Lambda})$ denotes the empirical Rademacher complexity of the robust loss class.*

The detailed proof is provided in Appendix E.

**Consistency and Interpretation.** The tightness of this bound is primarily governed by the empirical Rademacher complexity $\mathfrak{R}_n(\Psi_{\Theta, \Lambda})$. This term quantifies the capacity of the robust loss function class to fit random noise. For a detailed treatment of Rademacher complexity, we refer to (Bartlett & Mendelson, 2002). Regarding the convergence rate, it is well established that for standard parametric hypothesis spaces such as linear models or neural networks with bounded weights (Bartlett & Mendelson, 2002, see Lemma 22 and Theorem 18), the Rademacher complexity scales as $\mathcal{O}(C/\sqrt{n})$. Here $C$ represents a complexity constant that depends on the structural properties of the function class, such as parameter norms. Consequently, the complexity term naturally decays at a rate of $\mathcal{O}(1/\sqrt{n})$ as the sample size $n$ increases. Since the second term in the bound also scales as $\mathcal{O}(1/\sqrt{n})$, the overall generalization gap vanishes at a rate of $\mathcal{O}(1/\sqrt{n})$ as $n \to \infty$. This decay rate establishes the statistical consistency of our estimator.

While Section 3 addressed the computational challenge by deriving a tractable dual formulation for the nested ambiguity set, this uniform convergence bound ensures that the solution obtained from the empirical dual objective is a reliable estimator of the true worst-case risk.

# 5. Experiments

In this section, we empirically validate the proposed nested DRO framework through experiments on both synthetic data and the CIFAR-10 benchmark. We aim to demonstrate that our bilevel formulation effectively decouples geometric smoothing from statistical cleaning, thereby resolving the geometric conflation problem.

## 5.1. Robustness against Geometric Conflation

We begin by evaluating nested DRO against standard DRO, OR-WDRO (Nietert et al., 2023), and UOT-DRO (Wang et al., 2024) on a synthetic regression task. This task is specifically constructed so that informative data points are geometrically distant, while adversarial outliers are geometrically proximal but statistically anomalous.

**Experimental Setup.** We consider a linear regression task $y = \langle w^\star, x \rangle + b^\star + \xi$ in $\mathbb{R}^d$, with fixed true parameters $\|w^\star\|_2 = 2.0$ and $b^\star = 1.0$. The data distribution is a mixture of three distinct components parameterized as follows:

- Majority Normal (70%): The background data follows a uniform distribution $x \sim \text{Unif}[-2, 2]^d$ with standard Gaussian noise $\xi \sim \mathcal{N}(0, 1)$.

- Valid High-Leverage Points (10%): Distant inputs $x \sim \mathcal{N}(\mu, 0.2^2 \cdot I_d)$ with low noise $\xi \sim \mathcal{N}(0, 0.1^2)$, where $\mu = 10 \cdot \frac{w^\star}{\|w^\star\|}$.

- Hidden Outliers (20%): These samples are clustered near the geometric center ($x_{out} \sim \mathcal{N}(0, I_d)$) but are labeled adversarially. Specifically, the labels are generated by inverting the true signal and injecting a large bias shift: $y_{out} = \langle -w^\star, x \rangle + b^\star + 10 + \xi$, with noise $\xi \sim \mathcal{N}(0, 0.5^2)$.

A visualization of the dataset with $d = 1$ is illustrated in Figure 1a. We report the excess risk evaluated on the clean distribution, consisting of the normal and high-leverage components. Specifically, excess risk is defined as the difference between the mean absolute error of the learned model and that of an oracle model parameterized by the true coefficients $w^\star, b^\star$. The reported results in Figure 1b and 1c display the mean excess risk, with error bars representing the standard error of the mean. Full implementation and experimental details are provided in Appendix F.

**Results and Analysis.** Figure 1b illustrates the excess risk

*Table 2.* CIFAR-10 results under overlap-confounded geometric and statistical corruption. The clean-reference ERM model is trained on the matched clean split and is reported for reference only; all other methods are trained on the contaminated split. We report best clean, defined as the highest accuracy attained on the clean CIFAR-10 test set over the course of training.

| Method | Best clean (%) |
|---|---|
| Clean-reference ERM | 88.17 |
| Contaminated ERM | 75.00 |
| Nested DRO | **78.78** |
| Statistical-only trimming | 71.50 |
| Sequential robustify-then-trim | 74.65 |
| OR-WDRO | 70.11 |
| UOT-DRO | 74.10 |

as a function of sample size $N$ varying from 500 to 1000 with a fixed dimension $d = 5$. Despite the baselines benefiting from specific tuning, the results demonstrate that nested DRO consistently outperforms all competing methods. This empirical evidence confirms that our framework effectively decouples geometric smoothing from statistical cleaning, thereby successfully distinguishing proximal adversarial outliers from valid but distant high-leverage points.

We further investigated robustness against the curse of dimensionality by fixing $N = 500$ and varying the feature dimension $d$ from 10 to 50. This result shows that nested DRO consistently outperforms the baseline methods even as the dimensionality increases.

## 5.2. Performance on Corrupted CIFAR-10

To evaluate scalability on high-dimensional real-world data, we study nested DRO on CIFAR-10 using a ResNet-18 backbone under a mixed corruption regime that explicitly confounds geometric difficulty with statistical contamination. The training distribution is constructed by partitioning the CIFAR-10 training set into four groups. Specifically, 10% of the samples are assigned label-preserving hard rotations, 20% are corrupted by both hard rotations and label noise, and another 10% are corrupted by label noise only; the remaining 60% are left unchanged. This construction yields an overlap-confounded setting in which the learner must distinguish valid hard geometric examples from genuinely contaminated samples, while also handling cases where both forms of corruption co-exist in a single sample. We compare nested DRO with a clean-reference ERM model trained on the matched clean split, a contaminated ERM baseline trained on the corrupted split, the Statistics-only and Sequential robustify-then-trim ablations, and the coupled baselines OR-WDRO and UOT-DRO. Full implementation and experimental details are provided in Appendix F.

**Results and Analysis.** Table 2 reports the CIFAR-10 results

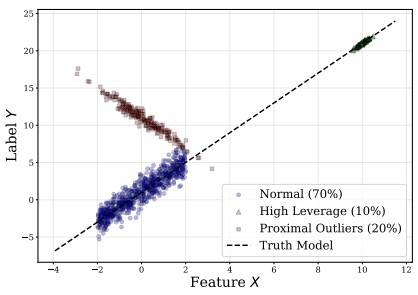

*(a)* Visualization of Geometric Conflation.

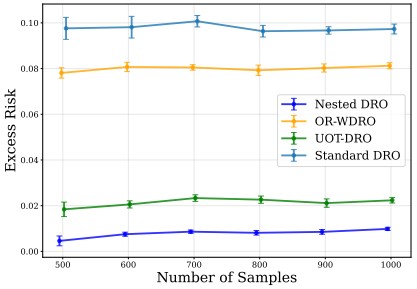

*(b)* Excess risk with various samples.

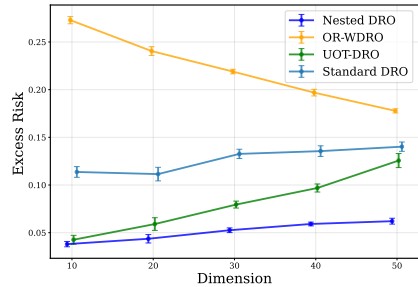

*(c)* Excess risk with various dimensions.

*Figure 1.* Visualization of geometric conflation and comparison of excess risk with respect to sample size and dimensionality. Unlike standard DRO which confuses high-leverage points with outliers, nested DRO successfully separates them using loss variance statistics, preserving the valid tail data.

under the overlap-confounded corruption protocol described above. We report the best clean test accuracy over training as a diagnostic robustness metric, uniformly across methods.

Several conclusions emerge from Table 2. First, contaminated ERM suffers a clear performance drop relative to the reference, confirming that the mixed corruption regime substantially degrades standard empirical risk minimization. Second, among the methods, nested DRO achieves the strongest clean-test performance, attaining $78.78\%$ best clean accuracy. Notably, several competing robust baselines are not only unable to improve upon nested DRO, but even underperform contaminated ERM itself. This pattern is consistent with the structure of our corruption protocol, where valid hard geometric examples and genuinely corrupted samples partially overlap. In this regime, trimming directly on per-sample cross-entropy tends to discard informative but difficult rotated samples together with true outliers, while trimming after geometric robustification can still suppress such valid hard examples because the robust surrogate remains large on them. Likewise, the adapted OR-WDRO and UOT-DRO baselines can over-entangle geometric difficulty with statistical contamination, leading to overly conservative reweighting that downweights high-leverage but valid samples rather than isolating true contaminants. These gains indicate that neither pure statistical filtering, nor a naive sequential combination of robustification and trimming, nor existing geometry–statistics coupled formulations are sufficient to match the proposed nested design. Overall, the results support the central claim of the paper: explicitly separating statistical cleaning from geometric robustification yields a stronger learning rule under overlap-confounded geometric and statistical corruption.

# 6. Conclusion

Existing DRO paradigms often suffer from geometric conflation, where valid high-leverage samples are indistinguishable from outliers due to the coupling of geometric and statistical constraints. In this paper, we address this by introducing nested DRO, a bilevel framework that strictly decouples geometric smoothing from statistical cleaning. Through duality analysis, we establish a rigorous theoretical equivalence between our nested formulation and variance regularization, providing a principled justification for variance-based cleaning heuristics. Our empirical results demonstrate that this decoupling effectively preserves informative data structure while rejecting heavy-tailed contamination, offering a robust path for learning under compound uncertainty. Future research includes extending the nested framework to a broader class of divergence constraints and investigating its application to sequential decision-making tasks.

# Acknowledgements

This work is supported by the National Key Research and Development Program of China under No. 2022YFA1004700, the National Natural Science Foundation of China under No. 72271187, and the Fundamental Research Funds for the Central Universities.

# Impact Statement

This paper presents work whose goal is to advance the field of Machine Learning, particularly regarding model robustness. There are many potential societal consequences of our work, none of which we feel must be specifically highlighted here.

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

# A. Proof of Theorem 3.1

We proceed by analyzing the nested optimization structure in (4) sequentially, starting from the inner geometric layer and then addressing the outer statistical layer.

**Step 1: Inner OT-constrained Maximization.** Consider the inner maximization problem:

$$\sup_{\mathbb{P}_W : W_c(\mathbb{P}_W, \mathbb{P}_D) \leq \rho} \mathbb{E}_{\mathbb{P}_W}[f_\theta(z)].$$

Since the ambiguity set $\mathcal{B}_W(\mathbb{P}_D, \rho) \triangleq \{\mathbb{P}_W : W_c(\mathbb{P}_W, \mathbb{P}_D) \leq \rho\}$ is convex and compact, strictly duality holds. We introduce a Lagrange multiplier $\lambda \geq 0$ for the transport cost constraint $W_c(\mathbb{P}_W, \mathbb{P}_D) \leq \rho$ and reformulate the primal problem as:

$$\sup_{\mathbb{P}_W} \inf_{\lambda \geq 0} \left\{ \mathbb{E}_{\mathbb{P}_W}[f_\theta(z)] - \lambda \big( W_c(\mathbb{P}_W, \mathbb{P}_D) - \rho \big) \right\}.$$

By exchanging the supremum and infimum, we obtain:

$$\inf_{\lambda \geq 0} \left\{ \lambda \rho + \sup_{\mathbb{P}_W} \big( \mathbb{E}_{\mathbb{P}_W}[f_\theta(z)] - \lambda W_c(\mathbb{P}_W, \mathbb{P}_D) \big) \right\}.$$

Using the Kantorovich-Rubinstein duality for the optimal transport term, the inner supremum over probability measures can be simplified to a pointwise maximization over the support of $\mathbb{P}_D$ (see Proposition 1 in (Sinha et al., 2017)). Specifically, the strong duality guarantees that:

$$\sup_{\mathbb{P}_W} \big( \mathbb{E}_{\mathbb{P}_W}[f_\theta(z)] - \lambda W_c(\mathbb{P}_W, \mathbb{P}_D) \big) = \mathbb{E}_{\mathbb{P}_D} \left[ \sup_{z' \in \mathcal{Z}} \{ f_\theta(z') - \lambda c(z', z) \} \right].$$

Defining the robust surrogate loss as $\psi_{\theta, \lambda}(z) \triangleq \sup_{z' \in \mathcal{Z}} \{ f_\theta(z') - \lambda c(z', z) \}$, the inner problem reduces to the tractable univariate minimization:

$$\inf_{\lambda \geq 0} \left\{ \lambda \rho + \mathbb{E}_{\mathbb{P}_D}[\psi_{\theta, \lambda}(z)] \right\}.$$

**Step 2: Outer $\phi$-Divergence Minimization.** Substituting the result from Step 1 into the original problem (4) yields

$$\inf_{\theta \in \Theta} \inf_{\mathbb{P}_D : \mathcal{D}_\phi(\mathbb{P}_D \| \mathbb{P}_0) \leq \epsilon} \inf_{\lambda \geq 0} \left\{ \lambda \rho + \mathbb{E}_{\mathbb{P}_D} \left[ \psi_{\theta, \lambda}(z) \right] \right\}.$$

Since the optimization over the intermediate distribution $\mathbb{P}_D$ and the dual variable $\lambda$ are both minimizations, we can interchange their order to isolate the inner divergence-constrained problem:

$$\inf_{\theta \in \Theta} \inf_{\lambda \geq 0} \left\{ \lambda \rho + \inf_{\mathbb{P}_D : \mathcal{D}_\phi(\mathbb{P}_D \| \mathbb{P}_0) \leq \epsilon} \mathbb{E}_{\mathbb{P}_D} \left[ \psi_{\theta, \lambda}(z) \right] \right\}. \tag{11}$$

We now focus on the inner minimization over $\mathbb{P}_D$ subject to the $\phi$-divergence constraint. Let $h(z) = \psi_{\theta, \lambda}(z)$ for brevity. The problem is to solve $\inf_{\mathbb{P}_D} \int h(z) d\mathbb{P}_D(z)$ subject to $\int \phi(\frac{d\mathbb{P}_D}{d\mathbb{P}_0}) d\mathbb{P}_0 \leq \epsilon$ and $\int d\mathbb{P}_D = 1$. By introducing the likelihood ratio $L(z) = \frac{d\mathbb{P}_D}{d\mathbb{P}_0}(z) \geq 0$, we formulate the Lagrangian with multipliers $\eta \geq 0$ (for the divergence constraint) and $\beta \in \mathbb{R}$ (for normalization):

$$\mathcal{L}(L, \eta, \beta) = \int h(z) L(z) d\mathbb{P}_0 + \eta \left( \int \phi(L(z)) d\mathbb{P}_0 - \epsilon \right) + \beta \left( \int L(z) d\mathbb{P}_0 - 1 \right).$$

Rearranging the terms, we aim to minimize the Lagrangian with respect to $L$:

$$\inf_{L \geq 0} \mathcal{L}(L, \eta, \beta) = -\eta \epsilon - \beta + \int \inf_{t \geq 0} \{ h(z) t + \eta \phi(t) + \beta t \} d\mathbb{P}_0(z).$$

We rewrite the integrand using the definition of the Fenchel conjugate $\phi^*(s) = \sup_t \{ st - \phi(t) \}$. Observe that:

$$\inf_{t \geq 0} \{ (h(z) + \beta) t + \eta \phi(t) \} = -\eta \sup_{t \geq 0} \left\{ \frac{-(h(z) + \beta)}{\eta} t - \phi(t) \right\} = -\eta \phi^* \left( \frac{-h(z) - \beta}{\eta} \right).$$

Thus, the dual problem becomes maximizing the infimum of the Lagrangian:

$$\sup_{\eta \geq 0, \beta} \left\{ -\eta\epsilon - \beta - \eta\mathbb{E}_{\mathbb{P}_0} \left[ \phi^* \left( \frac{-h(z) - \beta}{\eta} \right) \right] \right\}.$$

To match the standard form in (6), we apply a variable substitution $\mu = -\beta$. The maximization then becomes:

$$\sup_{\eta \geq 0, \mu} \left\{ \mu - \eta\epsilon - \eta\mathbb{E}_{\mathbb{P}_0} \left[ \phi^* \left( \frac{\mu - h(z)}{\eta} \right) \right] \right\}. \tag{12}$$

Note that the domain of the conjugate $\phi^*$ implies explicit constraints on the dual variables. Specifically, for the expectation to be finite, the argument $\frac{\mu - h(z)}{\eta}$ must lie within the effective domain of $\phi^*$. This imposes the constraint formulated in $\Lambda_{\phi, \psi_{\theta, \lambda}}$ in (8), which is consistent with the duality results and recession analysis presented in Theorem 5.3 of (Rahimian & Mehrotra, 2019).

**Step 3: Joint Formulation.** Combining the results from Step 1 and Step 2, we substitute (12) back into (11). Since the inner supremum over $(\eta, \mu)$ involves $\theta$ and $\lambda$ only through the scalar value $\psi_{\theta, \lambda}(z)$, we can merge the optimization operators. Restoring $h(z) = \psi_{\theta, \lambda}(z)$, we arrive at the final dual formulation:

$$\inf_{\theta \in \Theta, \lambda \geq 0} \sup_{(\eta, \mu) \in \Lambda_{\phi, \psi_{\theta, \lambda}}} \left\{ \lambda\rho + \mu - \eta\epsilon - \eta\,\mathbb{E}_{\mathbb{P}_0} \left[ \phi^* \left( \frac{\mu - \psi_{\theta, \lambda}(z)}{\eta} \right) \right] \right\}.$$

This completes the proof.

## B. Proof of Theorem 3.2 and Regularity Conditions

In this part we provide the detailed proof for Theorem 3.2 along with the complete characterization of the optimal worst case distributions. We begin by restating the proposition in its general form which includes the degenerate cases omitted from the main text for brevity.

**Proposition B.1.** *Let $\psi : \mathcal{Z} \to \mathbb{R}$ be a measurable function. Consider the distributionally robust optimization problem:*

$$\inf_{\mathbb{P}: \mathcal{D}_\phi(\mathbb{P} \| \mathbb{P}_0) \leq \epsilon} \mathbb{E}_{z \sim \mathbb{P}} \left[ \psi(z) \right].$$

*Suppose the radius satisfies $\epsilon > 0$. Then, Slater's condition holds, and strong duality obtains. Let $\eta^* \geq 0$ denote an optimal dual variable associated with the $\phi$-divergence constraint $\mathcal{D}_\phi(\mathbb{P} \| \mathbb{P}_0) \leq \epsilon$, consistent with the general dual formulation in Theorem 3.1. Then the optimal worst-case distribution $\mathbb{P}^*$ is characterized as follows:*

1. *Case $\phi(t) = \frac{1}{2}(t - 1)^2$ ($\chi^2$-divergence):*

   - *$\eta^* > 0$ : Assume the radius $\epsilon$ is sufficiently small such that the optimal density remains strictly positive on the support of $\mathbb{P}_0$. Then the optimal distribution is a linear reweighting of the base measure:*

   $$d\mathbb{P}^*(z) = \left( 1 - \frac{\psi(z) - \mathbb{E}_{\mathbb{P}_0}[\psi(z)]}{\eta^*} \right)_+ d\mathbb{P}_0(z).$$

   - *$\eta^* = 0$ : The optimal distribution is a singular measure supported on the global minimizers of $\psi(z)$. That is, $\mathbb{P}^*(\{z : \psi(z) = \inf_{z'} \psi(z')\}) = 1$.*

2. *Case $\phi(t) = t \log t - t + 1$ (KL-Divergence): Let $\psi^{\min} = \text{ess inf}_{\mathbb{P}_0} \psi(z)$ and assume the probability mass at the minimum is positive, i.e., $p^{\min} \triangleq \mathbb{P}_0(\psi(z) = \psi^{\min}) > 0$. Provided that the radius satisfies $0 < \epsilon < -\log p^{\min}$, then the optimal Lagrange multiplier satisfies $\eta^* > 0$ and the optimal distribution follows an exponential reweighting:*

   $$d\mathbb{P}^*(z) = \frac{e^{-\psi(z)/\eta^*}}{\mathbb{E}_{\mathbb{P}_0} \left[ e^{-\psi(z)/\eta^*} \right]} d\mathbb{P}_0(z).$$

*Proof.* Let $(\mathcal{Z}, \mathcal{F}, \mathbb{P}_0)$ be the underlying probability space. Since $\epsilon < \infty$, we restrict the search to probability measures $\mathbb{P}$ that are absolutely continuous with respect to the base measure $\mathbb{P}_0$ (denoted as $\mathbb{P} \ll \mathbb{P}_0$). By the Radon-Nikodym Theorem, there exists a unique measurable function $L : \mathcal{Z} \to [0, \infty)$, denoted as $L = \frac{d\mathbb{P}}{d\mathbb{P}_0}$, such that $d\mathbb{P}(z) = L(z)d\mathbb{P}_0(z)$.

This allows us to reformulate the primal optimization problem (4) in terms of the likelihood ratio $L$:

$$
\begin{aligned}
\inf_L \quad & \int_{\mathcal{Z}} \psi(z)L(z)d\mathbb{P}_0(z) \\
\text{s.t.} \quad & \int_{\mathcal{Z}} \phi(L(z))d\mathbb{P}_0(z) \leq \epsilon, \\
& \int_{\mathcal{Z}} L(z)d\mathbb{P}_0(z) = 1, \quad L(z) \geq 0 \text{ a.s.}
\end{aligned}
\tag{13}
$$

**Strong Duality.** The objective is linear in $L$, and the divergence constraint is convex in $L$ (since $\phi$ is convex). The Slater's condition requires the existence of a feasible point that strictly satisfies the inequality constraint. Consider the reference measure itself, i.e., $L_0(z) \equiv 1$. Since $\phi(1) = 0$, we have $\mathbb{E}_{\mathbb{P}_0}[\phi(1)] = 0 < \epsilon$. Thus, Slater's condition holds, guaranteeing strong duality. Let $\eta^* \geq 0$ and $\mu^* \in \mathbb{R}$ be the optimal Lagrange multipliers associated with the divergence constraint and the normalization constraint, respectively.

**Case $\chi^2$-Divergence.** Here $\phi(t) = \frac{1}{2}(t - 1)^2$. The constraint $\mathcal{D}_\phi(\mathbb{P}\|\mathbb{P}_0) \leq \epsilon$ implies $L \in L^2(\mathbb{P}_0)$. We formulate the Lagrangian functional:

$$
\mathcal{L}_{\chi^2}(L, \eta, \mu) = \int_{\mathcal{Z}} \psi L d\mathbb{P}_0 + \eta \left( \frac{1}{2} \int_{\mathcal{Z}} (L - 1)^2 d\mathbb{P}_0 - \epsilon \right) + \mu \left( \int_{\mathcal{Z}} L d\mathbb{P}_0 - 1 \right).
$$

*Subcase 1: $\eta^* > 0$.* We compute the first-order variation of $\mathcal{L}_{\chi^2}$ with respect to $L$ in the direction of an arbitrary test function $v$:

$$
\delta_L \mathcal{L}_{\chi^2}[v] = \int_{\mathcal{Z}} \left( \psi(z) + \eta^*(L(z) - 1) + \mu^* \right) v(z)d\mathbb{P}_0(z).
$$

Under the assumption that the radius $\epsilon$ is sufficiently small, the optimal density lies in the interior of the non-negative cone, i.e., $L^*(z) \geq 0$. For an interior solution, the necessary condition for optimality is that the first-order variation vanishes for all directions $v$. By the fundamental lemma of calculus of variations, this implies:

$$
\psi(z) + \eta^*(L^*(z) - 1) + \mu^* = 0.
$$

Rearranging this equation yields the explicit form for the optimal density:

$$
L^*(z) = 1 - \frac{\psi(z) + \mu^*}{\eta^*}.
$$

To determine $\mu^*$, we substitute this linear form into the normalization constraint $\int L^* d\mathbb{P}_0 = 1$:

$$
\int_{\mathcal{Z}} \left( 1 - \frac{\psi(z) + \mu^*}{\eta^*} \right) d\mathbb{P}_0(z) = 1 \implies 1 - \frac{\mathbb{E}_{\mathbb{P}_0}[\psi] + \mu^*}{\eta^*} = 1.
$$

This simplifies to $\mathbb{E}_{\mathbb{P}_0}[\psi] + \mu^* = 0$, yielding $\mu^* = -\mathbb{E}_{\mathbb{P}_0}[\psi]$. Substituting $\mu^*$ back gives the linear reweighting form:

$$
L^*(z) = 1 - \frac{\psi(z) - \mathbb{E}_{\mathbb{P}_0}[\psi(z)]}{\eta^*}.
$$

Since $L^*(z)$ must be nonnegative, we strictly enforce this constraint by applying the projection operator. This yields the finally reweighting form.

*Subcase 2: $\eta^* = 0$.* According to the KKT complementary slackness condition, we must have $\eta^* \cdot (\mathcal{D}_{\chi^2}(\mathbb{P}^*\|\mathbb{P}_0) - \epsilon) = 0$. The case $\eta^* = 0$ implies that the divergence constraint is slack, meaning the constraint boundary does not impede the minimization of the objective function. Consequently, the optimization effectively reduces to an unconstrained minimization

of the linear functional $\mathbb{E}_{\mathbb{P}}[\psi(z)]$ over the probability simplex. To see the structure of the solution, consider the first-order stationarity condition derived in Subcase 2.1. Setting $\eta^* \to 0$, the condition for $L^*(z) > 0$ degenerates to:

$$\psi(z) + \mu^* = 0 \implies \psi(z) = -\mu^*.$$

This equality implies that the optimal density $L^*(z)$ can only be non-zero on the level set where $\psi(z)$ is constant and equal to $-\mu^*$. To minimize the objective $\int \psi L d\mathbb{P}_0$, this constant level must correspond to the global infimum of $\psi$. Thus, the probability mass of $\mathbb{P}^*$ concentrates entirely on the set of global minimizers: $\mathbb{P}^*(\{z : \psi(z) = \inf_{z'} \psi(z')\}) = 1$.

**Case KL-Divergence.** Here $\phi(t) = t \log t - t + 1$. We first establish that the optimal dual variable satisfies $\eta^* > 0$ under the condition $0 < \epsilon < -\log p^{\min}$.

We proceed by contradiction. Assume that $\eta^* = 0$. By the KKT complementary slackness condition, $\eta^* \cdot (\mathcal{D}_{KL}(\mathbb{P}^* \| \mathbb{P}_0) - \epsilon) = 0$. If $\eta^* = 0$, the primal optimization effectively reduces to minimizing the linear functional $\mathbb{E}_{\mathbb{P}}[\psi(z)]$ over the probability simplex, ignoring the $\phi$-divergence constraint in the Lagrangian. The unconstrained minimizer of this linear objective concentrates all probability mass on the set of global minimizers of $\psi$, denoted by $\mathcal{Z}_{\min} \triangleq \{z : \psi(z) = \psi^{\min}\}$.

We assert that among all distributions supported on $\mathcal{Z}_{\min}$, the one closest to the base distribution $\mathbb{P}_0$ in terms of KL-Divergence is the conditional distribution of $\mathbb{P}_0$ restricted to $\mathcal{Z}_{\min}$, denoted as $\mathbb{P}_{\text{unc}}$. Its density ratio is given by

$$\frac{d\mathbb{P}_{\text{unc}}}{d\mathbb{P}_0}(z) = \begin{cases} \frac{1}{p^{\min}} & \text{if } z \in \mathcal{Z}_{\min}, \\ 0 & \text{otherwise.} \end{cases}$$

To justify this, consider any probability measure $\mathbb{Q}$ supported on $\mathcal{Z}_{\min}$, we have the decomposition:

$$\begin{aligned} \mathcal{D}_{KL}(\mathbb{Q} \| \mathbb{P}_0) &= \int_{\mathcal{Z}_{\min}} \log\left(\frac{d\mathbb{Q}}{d\mathbb{P}_0}\right) d\mathbb{Q} \\ &= \int_{\mathcal{Z}_{\min}} \log\left(\frac{d\mathbb{Q}}{d\mathbb{P}_{\text{unc}}} \cdot \frac{d\mathbb{P}_{\text{unc}}}{d\mathbb{P}_0}\right) d\mathbb{Q} \\ &= \int_{\mathcal{Z}_{\min}} \log\left(\frac{d\mathbb{Q}}{d\mathbb{P}_{\text{unc}}}\right) d\mathbb{Q} + \int_{\mathcal{Z}_{\min}} \log\left(\frac{1}{p^{\min}}\right) d\mathbb{Q} \\ &= \mathcal{D}_{KL}(\mathbb{Q} \| \mathbb{P}_{\text{unc}}) - \log p^{\min}. \end{aligned}$$

Since the KL-Divergence is non-negative, the minimum is uniquely attained when $\mathcal{D}_{KL}(\mathbb{Q} \| \mathbb{P}_{\text{unc}}) = 0$, i.e., $\mathbb{Q} = \mathbb{P}_{\text{unc}}$.

We then calculate the KL-Divergence of this best-case unconstrained candidate from $\mathbb{P}_0$:

$$\begin{aligned} \mathcal{D}_{KL}(\mathbb{P}_{\text{unc}} \| \mathbb{P}_0) &= \int_{\mathcal{Z}_{\min}} \log\left(\frac{1}{p^{\min}}\right) d\mathbb{P}_{\text{unc}}(z) \\ &= -\log p^{\min} \int_{\mathcal{Z}_{\min}} \frac{1}{p^{\min}} d\mathbb{P}_0(z) \\ &= -\log p^{\min}. \end{aligned}$$

This value represents the minimum divergence required to shift the probability mass entirely to the global minimizers of the loss. However, the hypothesis states that the trust region radius is strictly smaller than this distance, i.e., $\epsilon < -\log p^{\min}$. This implies $\mathcal{D}_{KL}(\mathbb{P}_{\text{unc}} \| \mathbb{P}_0) > \epsilon$. Thus, the unconstrained minimizer is infeasible with respect to the primal constraint $\mathcal{D}_{KL}(\mathbb{P} \| \mathbb{P}_0) \le \epsilon$. Consequently, the constraint must be active at the optimum, necessitating a strictly positive Lagrange multiplier, $\eta^* > 0$.

With $\eta^* > 0$ established, we consider the Lagrangian functional:

$$\mathcal{L}_{\text{KL}}(L, \eta, \mu) = \int_{\mathcal{Z}} \psi L d\mathbb{P}_0 + \eta\left(\int_{\mathcal{Z}} (L \log L - L + 1) d\mathbb{P}_0 - \epsilon\right) + \mu\left(\int_{\mathcal{Z}} L d\mathbb{P}_0 - 1\right).$$

The stationarity condition with respect to $L$ yields:

$$\psi(z) + \eta^*(\log L^*(z) + 1 - 1) + \mu^* = 0.$$

Rearranging the terms:

$$\log L^*(z) = -\frac{\psi(z) + \mu^*}{\eta^*} \implies L^*(z) = e^{-\frac{\psi(z)+\mu^*}{\eta^*}}.$$

Note that the exponential function guarantees $L^*(z) > 0$, satisfying the non-negativity constraint automatically. We determine $\mu^*$ via the normalization constraint:

$$\int_{\mathcal{Z}} e^{-\psi(z)/\eta^*} e^{-\mu^*/\eta^*} d\mathbb{P}_0(z) = 1.$$

Solving for the constant term involving $\mu^*$ yields

$$e^{\mu^*/\eta^*} = \int_{\mathcal{Z}} e^{-\psi(z)/\eta^*} d\mathbb{P}_0(z) = \mathbb{E}_{\mathbb{P}_0}\left[e^{-\psi(z)/\eta^*}\right].$$

Substituting this back into the expression for $L^*(z)$ yields the optimal density:

$$L^*(z) = \frac{e^{-\psi(z)/\eta^*}}{e^{\mu^*/\eta^*}} = \frac{e^{-\psi(z)/\eta^*}}{\mathbb{E}_{\mathbb{P}_0}\left[e^{-\psi(z)/\eta^*}\right]},$$

which corresponds to the exponential reweighting form stated in the proposition. □

## C. Proof of Theorem 3.3

Set $\phi(t) = \frac{1}{2}(t-1)^2$. Its Fenchel conjugate is

$$\phi^*(s) = \begin{cases} \frac{1}{2}s^2 + s, & s \geq -1 \\ -\frac{1}{2}, & s < -1 \end{cases}.$$

We focus our analysis on the regime where the uncertainty radius $\epsilon$ is sufficiently small such that the optimal argument of the conjugate function falls strictly within its quadratic domain. Fix $(\theta, \lambda)$. Then from theorem 3.1, the corresponding dual problem of $V_{\chi^2}(\mathbb{P}_0; \rho, \epsilon)$ is

$$\inf_{\theta \in \Theta, \lambda \geq 0} \sup_{(\eta, \mu) \in \Lambda_{\chi^2, \psi_{\theta,\lambda}}} \left\{ \lambda\rho + \mu - \eta\epsilon - \eta\, \mathbb{E}_{\mathbb{P}_0}\left[\frac{\mu - \psi_{\theta,\lambda}(z)}{\eta} + \frac{1}{2}\left(\frac{\mu - \psi_{\theta,\lambda}(z)}{\eta}\right)^2\right] \right\}, \tag{14}$$

where the feasible set $\Lambda_{\chi^2, \psi_{\theta,\lambda}}$ is defined in (8) with $\psi_{\theta,\lambda}$ in place of $h$ and $\phi(t) = \frac{1}{2}(t-1)^2$. The associated inner maximization is

$$\sup_{(\eta, \mu) \in \Lambda_{\chi^2, \psi_{\theta,\lambda}}} \left\{ S_{\chi^2}(\eta, \mu) \triangleq \mu - \eta\epsilon - \eta\, \mathbb{E}_{\mathbb{P}_0}\left[\frac{\mu - \psi_{\theta,\lambda}(z)}{\eta} + \frac{1}{2}\left(\frac{\mu - \psi_{\theta,\lambda}(z)}{\eta}\right)^2\right] \right\}. \tag{15}$$

We then solve (15) by partitioning the feasible set into the two cases $\eta = 0$ and $\eta > 0$.

**Case 1: $\eta = 0$.** If $\eta = 0$, the feasibility condition in (8) requires $\mu \leq \psi_{\theta,\lambda}(z)$ for all $z \in \mathcal{Z}$. However, the quadratic validity condition implies that as $\eta \to 0$, we must have $\mu - \psi_{\theta,\lambda}(z) \geq 0$, i.e., $\mu \geq \psi_{\theta,\lambda}(z)$. Combining these inequalities yields $\psi_{\theta,\lambda}(z) \leq \mu \leq \psi_{\theta,\lambda}(z)$, which implies that $\psi_{\theta,\lambda}(z)$ is almost surely constant. Excluding this trivial case, the condition $\eta = 0$ contradicts the quadratic validity assumption. Thus, we must have $\eta > 0$.

**Case 2: $\eta > 0$.** Since $\lim_{t \to 0} \frac{\phi(t)}{t} = +\infty$, no further restriction applies to $\mu$, i.e., $\mu \in \mathbb{R}$. Since $\eta > 0$ and the quadratic form applies, we substitute $\phi^*(s) = \frac{1}{2}s^2 + s$ into the objective:

$$S_{\chi^2}(\eta, \mu) = \mathbb{E}_{\mathbb{P}_0}[\psi_{\theta,\lambda}] - \eta\epsilon - \frac{1}{2\eta}\left(\mathrm{Var}_{\mathbb{P}_0}(\psi_{\theta,\lambda}) + (\mu - \mathbb{E}_{\mathbb{P}_0}[\psi_{\theta,\lambda}])^2\right). \tag{16}$$

For fixed $\eta > 0$, (16) is concave in $\mu$ and maximized at $\mu^* = \mathbb{E}_{\mathbb{P}_0}[\psi_{\theta,\lambda}]$. Substituting $\mu^*$ back yields:

$$\sup_{\mu} S_{\chi^2}(\eta, \mu) = \mathbb{E}_{\mathbb{P}_0}[\psi_{\theta,\lambda}] - \eta\epsilon - \frac{\mathrm{Var}_{\mathbb{P}_0}(\psi_{\theta,\lambda})}{2\eta}. \tag{17}$$

Maximizing this over $\eta > 0$ gives the stationary point $\eta^* = \sqrt{\frac{\text{Var}(\psi_{\theta,\lambda})}{2\epsilon}}$, yielding

$$\sup_{\eta>0,\mu} S_{\chi^2}(\eta,\mu) = \mathbb{E}_{\mathbb{P}_0}[\psi_{\theta,\lambda}] - \sqrt{2\epsilon \, \text{Var}_{\mathbb{P}_0}(\psi_{\theta,\lambda})}.$$

And substituting this expression back to (14) yields the desired result.

## D. Derivations and Asymptotic Analysis for KL-Divergence

In this appendix, we provide the detailed derivations for the KL-divergence case mentioned in Remark 3.4. We first derive the exact dual formulation. Then, to address the computational intractability of the inner maximization, we establish a general mathematical proposition regarding the asymptotic behavior of logarithmic moment generating functions and apply it to derive the final asymptotic variance-regularized objective.

**Theorem D.1** (KL-Divergence Dual). *Let $V_{\text{KL}}(\mathbb{P}_0; \rho, \epsilon)$ denote the objective $V_\phi(\mathbb{P}_0; \rho, \epsilon)$ in (4), where the divergence function is specified as the KL-Divergence $\phi(t) = t \log t - t + 1$. Under the conditions of Theorem 3.1, the objective admits the following dual representation:*

$$V_{\text{KL}}(\mathbb{P}_0; \rho, \epsilon) = \inf_{\theta \in \Theta, \, \lambda \geq 0} \left\{ \lambda\rho + \max\left( \sup_{\eta>0} F_{\theta,\lambda}(\eta; \epsilon), \, \inf_{z \in \mathcal{Z}} \psi_{\theta,\lambda}(z) \right) \right\}, \tag{18}$$

*where $\psi_{\theta,\lambda}(z)$ is defined in (7) and $F_{\theta,\lambda}(\eta; \epsilon) \triangleq -\eta\epsilon - \eta \log \mathbb{E}_{\mathbb{P}_0}[e^{-\psi_{\theta,\lambda}(z)/\eta}]$.*

*Proof.* Set $\phi(t) = t \log t - t + 1$. Its Fenchel conjugate is $\phi^*(x) = e^x - 1$, and

$$\lim_{t \to 0} \frac{\phi(t)}{t} = \infty. \tag{19}$$

Fix $(\theta, \lambda)$. From theorem 3.1, the corresponding dual problem of $V_{\text{KL}}(\mathbb{P}_0; \rho, \epsilon)$ is

$$\inf_{\theta \in \Theta, \lambda \geq 0} \sup_{(\eta,\mu) \in \Lambda_{\text{KL}, \psi_{\theta,\lambda}}} \left\{ \lambda\rho + \mu - \eta\epsilon - \eta\mathbb{E}_{\mathbb{P}_0}\left[ \exp\left( \frac{\mu - \psi_{\theta,\lambda}(z)}{\eta} \right) - 1 \right] \right\},$$

where the feasible set $\Lambda_{\text{KL}, \psi_{\theta,\lambda}}$ is defined in (8) with $\psi_{\theta,\lambda}$ in place of $h$ and $\phi(t) = t \log t - t + 1$. The associated inner maximization is

$$\sup_{(\eta,\mu) \in \Lambda_{\text{KL}, \psi_{\theta,\lambda}}} \left\{ S_{\text{KL}}(\eta,\mu) \triangleq \mu - \eta\epsilon - \eta\mathbb{E}_{\mathbb{P}_0}\left[ \exp\left( \frac{\mu - \psi_{\theta,\lambda}(z)}{\eta} \right) - 1 \right] \right\}.$$

Since (19) holds for both the $\chi^2$ and KL-Divergences, we follow the approach of Theorem 3.3 and solve (D) by partitioning the feasible set into two subsets, $\{(\eta,\mu) \mid \eta > 0, \, \mu \in \mathbb{R}\}$ and $\{(\eta,\mu) \mid \eta = 0, \, \mu \leq \psi_{\theta,\lambda}(z), \, \forall z \in \mathcal{Z}\}$, and then evaluating the inner objective on each subset to determine the maximal value.

**Case 1:** $\eta = 0$ and $\mu \leq \psi_{\theta,\lambda}(z), \, \forall z \in \mathcal{Z}$. The analysis in this case matches Case 1 in Theorem 3.3. Maximizing $S_{\text{KL}}(\eta,\mu)$ under this constraint pushes $\mu$ up to the tightest feasible value. This yields

$$\sup_{\eta=0, \, \mu \leq \psi_{\theta,\lambda}(z)} S_{\text{KL}}(\eta,\mu) = \inf_z \psi_{\theta,\lambda}(z).$$

**Case 2:** $\eta > 0$ and $\mu \in \mathbb{R}$. We first maximize with respect to $\mu$. Taking the derivative of $S_{\text{KL}}$ with respect to $\mu$ yields

$$\frac{\partial S_{\text{KL}}}{\partial \mu} = 1 - \mathbb{E}_{\mathbb{P}_0}\left[ \exp\left( \frac{\mu - \psi_{\theta,\lambda}(z)}{\eta} \right) \right].$$

Setting the derivative to zero yields the stationary condition

$$\mathbb{E}_{z \sim \mathbb{P}_0}\left[ \exp\left( \frac{\mu - \psi_{\theta,\lambda}(z)}{\eta} \right) \right] = e^{\mu/\eta} \, \mathbb{E}_{\mathbb{P}_0}[e^{-\psi_{\theta,\lambda}(z)/\eta}] = 1.$$

Since the second derivative

$$\frac{\partial^2 S_{\mathrm{KL}}}{\partial \mu^2} = -\frac{1}{\eta} \mathbb{E}_{\mathbb{P}_0}\Big[\exp\Big(\frac{\mu - \psi_{\theta,\lambda}(z)}{\eta}\Big)\Big]$$

is strictly negative for all $\eta > 0$, the function $S_{\mathrm{KL}}$ is concave in $\mu$. Therefore, the stationary point corresponds to the global maximizer, which is given by

$$\mu^* = \eta \, \log\Big(\frac{1}{\mathbb{E}_{\mathbb{P}_0}[e^{-\psi_{\theta,\lambda}(z)/\eta}]}\Big).$$

Substituting this expression into $S_{\mathrm{KL}}(\eta, \mu)$ yields

$$S_{\mathrm{KL}}(\eta, \mu^*) = -\eta\epsilon - \eta \log \mathbb{E}_{\mathbb{P}_0}[e^{-\psi_{\theta,\lambda}(z)/\eta}] = F_{\theta,\lambda}(\eta; \epsilon).$$

Maximizing the above expression with respect to $\eta > 0$ then produces $\sup_{\eta>0} F_{\theta,\lambda}(\eta; \epsilon)$. Combining this result with (D) in Case 1 gives precisely the inner supremum term appearing in (18). Then substituting the inner supremum term into the outer minimization over $(\theta, \lambda)$ completes the proof of (18). $\qquad\square$

The exact dual formulation derived in Theorem D.1 involves an implicit maximization that generally lacks a closed-form solution, hindering direct implementation. To address this computational challenge, we establish a general asymptotic property for functions of the form $-\eta\epsilon - \eta \log \mathbb{E}_{\mathbb{P}_0}[e^{-\psi/\eta}]$, which appear centrally in the KL-dual formulation.

**Proposition D.2.** *Consider the function $F_\epsilon : \mathbb{R}_+ \to \mathbb{R}$ defined by*

$$F_\epsilon(\eta) \triangleq -\eta\epsilon - \eta \log \mathbb{E}_{\mathbb{P}_0}\Big[e^{-\psi(z)/\eta}\Big], \tag{20}$$

*where $\psi : \mathcal{Z} \to [0, \infty)$ is a $\mathbb{P}_0$-integrable function that is not almost surely constant.*

*(i) Suppose that there exists a constant $M > 0$ such that the moment generating function of $\psi$ is finite on $(0, 1/M]$, i.e.,*

$$\mathbb{E}_{\mathbb{P}_0}[e^{t\psi(z)}] < \infty, \quad \forall t \in (0, 1/M].$$

*Then for all $\eta \geq M$,*

$$F_\epsilon(\eta) = \widetilde{F}_\epsilon(\eta) + R(\eta), \tag{21}$$

*where*

$$\widetilde{F}_\epsilon(\eta) = \mathbb{E}_{\mathbb{P}_0}[\psi(z)] - \eta\epsilon - \frac{\mathrm{Var}_{\mathbb{P}_0}(\psi(z))}{2\eta}, \tag{22}$$

*and the remainder term satisfies $R(\eta) = \mathcal{O}(\eta^{-2})$. Moreover, there exist constants $K_0 > 0$ and $\eta_0 \geq M$ such that for all $\eta \geq \eta_0$,*

$$|R(\eta)| \leq \frac{K_0}{\eta^2}, \qquad |R'(\eta)| \leq \frac{K_0}{\eta^3}. \tag{23}$$

*(ii) Let $\psi^{\min} = \operatorname{ess\,inf}_{\mathbb{P}_0} \psi(z)$ and assume that the probability mass at the minimum is positive, i.e.,*

$$p^{\min} \triangleq \mathbb{P}_0(\psi(z) = \psi^{\min}) > 0.$$

*Then, it follows that $p^{\min} < 1$, and for all $0 < \epsilon < -\log p^{\min}$, the function $F_\epsilon$ is strictly concave on $(0, \infty)$ and admits a unique maximizer $\eta^* \in (0, \infty)$.*

*(iii) Under the above conditions, define*

$$\epsilon_0 \triangleq \frac{\mathrm{Var}_{\mathbb{P}_0}(\psi(z))}{2(\eta_1)^2} - \frac{K_0}{(\eta_1)^3}, \quad \text{with} \quad \eta_1 \triangleq \max\Big\{\eta_0, \frac{4K_0}{\mathrm{Var}_{\mathbb{P}_0}(\psi(z))}\Big\}. \tag{24}$$

*Then for all $0 < \epsilon \leq \min\{-\frac{1}{2}\log p^{\min}, \epsilon_0\}$,*

$$\sup_{\eta>0} F_\epsilon(\eta) = \mathbb{E}_{\mathbb{P}_0}[\psi(z)] - \sqrt{2\epsilon \, \mathrm{Var}_{\mathbb{P}_0}(\psi(z))} + \mathcal{O}\Big(\frac{\epsilon}{\mathrm{Var}_{\mathbb{P}_0}(\psi(z))}\Big). \tag{25}$$

*Proof.* (i) Define the centered variable $\psi_c = \psi - \mathbb{E}_{\mathbb{P}_0}[\psi(z)]$. Then the function $L(\eta) \triangleq \log \mathbb{E}_{\mathbb{P}_0}\left[e^{-\psi(z)/\eta}\right]$ can be decomposed as

$$L(\eta) = -\frac{\mathbb{E}_{\mathbb{P}_0}[\psi(z)]}{\eta} + \log \mathbb{E}_{\mathbb{P}_0}\left[e^{-\psi_c(z)/\eta}\right] = -\frac{\mathbb{E}_{\mathbb{P}_0}[\psi(z)]}{\eta} + K_{\psi_c}\left(-\frac{1}{\eta}\right),$$

where $K_{\psi_c}(t) \triangleq \log \mathbb{E}_{\mathbb{P}_0}\left[e^{t\psi_c(z)}\right]$ is the cumulant generating function (CGF) of $\psi_c$.

Since $\mathbb{E}_{\mathbb{P}_0}[e^{t\psi(x)}] < \infty$ for all $t \in (0, 1/M]$ and $\psi(z) \geq 0$, we obtain $K_{\psi_c}(t)$ is finite for all $t \in [-1/M, 1/M]$. Therefore, it is analytic in a neighborhood of $t = 0$. This analyticity guarantees the Maclaurin series expansion in terms of the cumulants $\kappa_k$:

$$K_{\psi_c}(t) = \sum_{k=1}^{\infty} \frac{\kappa_k}{k!} t^k, \quad t \in \left[-\frac{1}{M}, \frac{1}{M}\right]$$

where

$$\kappa_k \triangleq \left.\frac{d^k}{dt^k} K_{\psi_c}(t)\right|_{t=0}.$$

For example, the first two cumulants are $\kappa_1 = \mathbb{E}_{\mathbb{P}_0}[\psi_c] = 0$ and $\kappa_2 = \mathrm{Var}_{\mathbb{P}_0}(\psi_c) = \mathrm{Var}_{\mathbb{P}_0}(\psi)$.

Substituting $t = -1/\eta$ into the series, and using $\kappa_1 = 0$, we know that for $\eta \geq M$,

$$L(\eta) = -\frac{\mathbb{E}_{\mathbb{P}_0}[\psi(z)]}{\eta} + \sum_{k=2}^{\infty} \frac{\kappa_k}{k!}\left(-\frac{1}{\eta}\right)^k. \tag{26}$$

Expanding the first few terms, we have

$$L(\eta) = -\frac{\mathbb{E}_{\mathbb{P}_0}[\psi(z)]}{\eta} + \frac{\mathrm{Var}_{\mathbb{P}_0}(\psi)}{2\eta^2} + \mathcal{O}\left(\frac{1}{\eta^3}\right).$$

Consequently, the expansion for $F_\epsilon(\eta) = -\eta\epsilon - \eta L(\eta)$ is

$$F_\epsilon(\eta) = \mathbb{E}_{\mathbb{P}_0}[\psi(z)] - \eta\epsilon - \frac{\mathrm{Var}_{\mathbb{P}_0}(\psi)}{2\eta} + \mathcal{O}\left(\frac{1}{\eta^2}\right),$$

which gives (21). And exactly

$$R(\eta) = -\eta \sum_{k=3}^{\infty} \frac{\kappa_k}{k!}\left(-\frac{1}{\eta}\right)^k = \sum_{k=3}^{\infty} \frac{\kappa_k}{k!}(-1)^{k+1}\eta^{-(k-1)}.$$

To control the derivative, differentiate the series termwise (which is valid due to the analyticity of $K_{\psi_c}$):

$$R'(\eta) = \sum_{k=3}^{\infty} \frac{\kappa_k}{k!}(-1)^{k+1}(-(k-1))\,\eta^{-k}.$$

Therefore $R'(\eta) = \mathcal{O}(\frac{1}{\eta^3})$. And hence there exists $K_0 > 0$ and $\eta_0 \geq M$ such that (23) holds.

(ii) Since $\psi(z)$ is not almost surely constant with respect to $\mathbb{P}_0$, the probability mass at its essential infimum must be strictly less than one, i.e., $p^{\min} < 1$. Then will show that $F_\epsilon(\eta)$ is strictly concave with respect to $\eta$ on $(0, \infty)$ and, by analyzing the limits of $F'_\epsilon(\eta)$ at the boundaries, demonstrate that a unique stationary point $\eta^* \in (0, \infty)$ exists, which is therefore the unique global maximizer.

Define

$$M(\eta) \triangleq \mathbb{E}_{\mathbb{P}_0}\left[e^{-\psi(z)/\eta}\right]. \tag{27}$$

Then $F_\epsilon(\eta) = -\eta\epsilon - \eta \log M(\eta)$. Differentiating $F_\epsilon$ with respect to $\eta$ gives

$$F'_\epsilon(\eta) = -\epsilon - \log M(\eta) - \eta\frac{M'(\eta)}{M(\eta)}.$$

Note that

$$M'(\eta) = \mathbb{E}_{\mathbb{P}_0}\left[\frac{\partial}{\partial \eta} e^{-\psi(z)/\eta}\right] = \mathbb{E}_{\mathbb{P}_0}\left[e^{-\psi(z)/\eta}\frac{\psi(z)}{\eta^2}\right].$$

Substituting this result into the expression for $F'_\epsilon(\eta)$ yields

$$F'_\epsilon(\eta) = -\epsilon - \log M(\eta) - \frac{1}{\eta}\frac{\mathbb{E}_{\mathbb{P}_0}\left[e^{-\psi(z)/\eta}\,\psi(z)\right]}{\mathbb{E}_{\mathbb{P}_0}\left[e^{-\psi(z)/\eta}\right]}.$$

To simplify the expectation ratio appearing in $F'_\epsilon(\eta)$, we define a new probability measure $\mathbb{Q}_\eta$ that reweights $\mathbb{P}_0$ proportionally to the exponential term in $M(\eta)$:

$$d\mathbb{Q}_\eta(z) \triangleq \frac{e^{-\psi(z)/\eta}}{M(\eta)}\,d\mathbb{P}_0(z). \tag{28}$$

This ensures $\mathbb{E}_{\mathbb{Q}_\eta}[1] = 1$. Then

$$\mathbb{E}_{\mathbb{Q}_\eta}[\psi(z)] = \frac{\mathbb{E}_{\mathbb{P}_0}\left[e^{-\psi(z)/\eta}\,\psi(z)\right]}{\mathbb{E}_{\mathbb{P}_0}\left[e^{-\psi(z)/\eta}\right]} = \eta^2 \frac{M'(\eta)}{M(\eta)}. \tag{29}$$

This representation allows $F'_\epsilon(\eta)$ to be written as

$$F'_\epsilon(\eta) = -\epsilon - \log M(\eta) - \frac{\mathbb{E}_{\mathbb{Q}_\eta}[\psi(z)]}{\eta}. \tag{30}$$

Then

$$F''_\epsilon(\eta) = -\frac{M'(\eta)}{M(\eta)} + \frac{\mathbb{E}_{\mathbb{Q}_\eta}[\psi]}{\eta^2} - \frac{1}{\eta}\frac{d}{d\eta}\mathbb{E}_{\mathbb{Q}_\eta}[\psi]. \tag{31}$$

From (29), we know that

$$\begin{aligned}
\frac{d}{d\eta}\mathbb{E}_{\mathbb{Q}_\eta}[\psi] &= \frac{d}{d\eta}\left(\frac{\mathbb{E}_{\mathbb{P}_0}[\psi\, e^{-\psi/\eta}]}{M(\eta)}\right) = \frac{M(\eta)\,\mathbb{E}_{\mathbb{P}_0}[\psi^2 e^{-\psi/\eta}] - \left(\mathbb{E}_{\mathbb{P}_0}[\psi\, e^{-\psi/\eta}]\right)^2}{M^2(\eta)\eta^2} \\
&= \frac{1}{\eta^2}\left(\mathbb{E}_{\mathbb{Q}_\eta}[\psi^2] - \left(\mathbb{E}_{\mathbb{Q}_\eta}[\psi]\right)^2\right) = \frac{\text{Var}_{\mathbb{Q}_\eta}(\psi)}{\eta^2}.
\end{aligned} \tag{32}$$

Substituting (29) and (32) into (31) gives

$$F''_\epsilon(\eta) = -\frac{M'(\eta)}{M(\eta)} + \frac{\eta^2 M'(\eta)/M(\eta)}{\eta^2} - \frac{1}{\eta}\cdot\frac{\text{Var}_{\mathbb{Q}_\eta}(\psi)}{\eta^2} = -\frac{\text{Var}_{\mathbb{Q}_\eta}(\psi)}{\eta^3} < 0. \tag{33}$$

The strict inequality follows from the fact that $\text{Var}_{\mathbb{Q}_\eta}(\psi) > 0$ for all $\eta > 0$. Indeed, by the definition in (28), $\mathbb{Q}_\eta$ is absolutely continuous with respect to $\mathbb{P}_0$ with a strictly positive density on the support of $\mathbb{P}_0$. Since $\psi$ is not almost surely constant under $\mathbb{P}_0$, it is also not almost surely constant under $\mathbb{Q}_\eta$, implying $\text{Var}_{\mathbb{Q}_\eta}(\psi) > 0$. Hence, $F_\epsilon(\eta)$ is strictly concave on $(0, \infty)$.

The strict concavity implies that if a stationary point $\eta^* \in (0, \infty)$ satisfying $F'(\eta^*) = 0$ exists, it must be the unique global maximizer on $(0, \infty)$. To prove existence, we analyze the limits of $F'_\epsilon(\eta)$ at the boundaries 0 and $\infty$.

**We first analyze the limit as $\eta \to \infty$.** Since $\psi(z) \geq 0$, we have $0 \leq e^{-\psi(z)/\eta} \leq 1$ and $-\psi(z)/\eta \to 0$ pointwise as $\eta \to \infty$. By the dominated convergence theorem,

$$\lim_{\eta\to\infty} M(\eta) = \lim_{\eta\to\infty} \mathbb{E}_{\mathbb{P}_0}\left[e^{-\psi(z)/\eta}\right] = \mathbb{E}_{\mathbb{P}_0}[1] = 1.$$

Since $\mathbb{E}_{\mathbb{P}_0}[\psi(z)] < \infty$, then again by dominated convergence,

$$\lim_{\eta\to\infty}\mathbb{E}_{\mathbb{Q}_\eta}[\psi(z)] = \lim_{\eta\to\infty}\frac{\mathbb{E}_{\mathbb{P}_0}[\psi(z)\, e^{-\psi(z)/\eta}]}{M(\eta)} = \frac{\mathbb{E}_{\mathbb{P}_0}[\psi(z)]}{1} = \mathbb{E}_{\mathbb{P}_0}[\psi(z)].$$

Therefore

$$\lim_{\eta \to \infty} \frac{1}{\eta} \mathbb{E}_{\mathbb{Q}_\eta}[\psi(z)] = 0.$$

Putting these limits together gives

$$\lim_{\eta \to \infty} F'_\epsilon(\eta) = -\epsilon - 0 - 0 = -\epsilon < 0. \tag{34}$$

**We then analyze the limit as** $\eta \to 0^+$. Since $\psi(z) \geq 0$, it follows that $\psi^{\min} \geq 0$. We first analyze $\log M(\eta)$. Since

$$M(\eta) = e^{-\psi^{\min}/\eta} \mathbb{E}_{\mathbb{P}_0}\big[e^{-(\psi(z)-\psi^{\min})/\eta}\big],$$

we have

$$\log M(\eta) = -\frac{\psi^{\min}}{\eta} + \log \mathbb{E}_{\mathbb{P}_0}\big[e^{-(\psi(z)-\psi^{\min})/\eta}\big].$$

Hence,

$$F'_\epsilon(\eta) = -\epsilon - \log M(\eta) - \frac{1}{\eta} \mathbb{E}_{\mathbb{Q}_\eta}[\psi(z)]$$

$$= -\epsilon - \log \mathbb{E}_{\mathbb{P}_0}\big[e^{-(\psi(z)-\psi^{\min})/\eta}\big] + \frac{\psi^{\min}}{\eta} - \frac{1}{\eta} \mathbb{E}_{\mathbb{Q}_\eta}[\psi(z)].$$

We now justify the limit of the last two terms as $\eta \to 0^+$. Define $h(z) \triangleq \psi(z) - \psi^{\min} \geq 0$. By the definition of $\mathbb{Q}_\eta$, we can write

$$\mathbb{E}_{\mathbb{Q}_\eta}[\psi] = \frac{\mathbb{E}_{\mathbb{P}_0}[\psi(z)e^{-\psi(z)/\eta}]}{\mathbb{E}_{\mathbb{P}_0}[e^{-\psi(z)/\eta}]} = \frac{\mathbb{E}_{\mathbb{P}_0}[(\psi^{\min}+h(z))e^{-h(z)/\eta}]}{\mathbb{E}_{\mathbb{P}_0}[e^{-h(z)/\eta}]} = \psi^{\min} + \frac{\mathbb{E}_{\mathbb{P}_0}[h(z)e^{-h(z)/\eta}]}{\mathbb{E}_{\mathbb{P}_0}[e^{-h(z)/\eta}]}.$$

Hence,

$$\frac{\psi^{\min}}{\eta} - \frac{1}{\eta} \mathbb{E}_{\mathbb{Q}_\eta}[\psi] = -\frac{1}{\eta} \cdot \frac{\mathbb{E}_{\mathbb{P}_0}[h(z)e^{-h(z)/\eta}]}{\mathbb{E}_{\mathbb{P}_0}[e^{-h(z)/\eta}]}. \tag{35}$$

We claim that the right-hand side of (35) converges to 0 as $\eta \to 0^+$. Observe that

$$\lim_{\eta \to 0^+} \mathbb{E}_{\mathbb{P}_0}[e^{-h(z)/\eta}] = \mathbb{P}_0(h(z) = 0) = \mathbb{P}_0(\psi(z) = \psi^{\min}) = p^{\min} > 0. \tag{36}$$

Therefore, to show that the right-hand side of (35) vanishes, it suffices to show

$$\lim_{\eta \to 0^+} \frac{\mathbb{E}_{\mathbb{P}_0}[h(z)e^{-h(z)/\eta}]}{\eta} = 0.$$

For $\eta > 0$, define the nonnegative function

$$f_\eta(z) \triangleq \frac{h(z)e^{-h(z)/\eta}}{\eta} \geq 0.$$

We first establish pointwise convergence. If $h(z) = 0$, then $f_\eta(z) = 0$ for all $\eta > 0$. If $h(z) > 0$, then as $\eta \to 0^+$, the exponential term $e^{-h(z)/\eta}$ decays faster than the linear factor $\frac{h(z)}{\eta}$ grows, so $f_\eta(z) \to 0$. Thus $f_\eta(z) \to 0$ as $\eta \to 0^+$ for all $z$.

Next, we provide an $\eta$-independent integrable envelope to apply the dominated convergence theorem. For fixed $\eta > 0$, consider the scalar function $g_\eta(u) \triangleq ue^{-u/\eta}$ on $u \geq 0$. Then

$$g'_\eta(u) = e^{-u/\eta}\Big(1 - \frac{u}{\eta}\Big), \qquad g'_\eta(u) = 0 \iff u = \eta,$$

and $g_\eta$ attains its maximum at $u = \eta$ with $g_\eta(\eta) = \eta e^{-1}$. Therefore, for all $u \geq 0$,

$$0 \leq ue^{-u/\eta} \leq \eta e^{-1}.$$

Applying this with $u = h(z)$ yields

$$0 \leq f_\eta(z) = \frac{h(z)e^{-h(z)/\eta}}{\eta} \leq e^{-1} \qquad \text{for all } z \text{ and all } \eta > 0.$$

The constant $e^{-1}$ is integrable with respect to $\mathbb{P}_0$ since $\mathbb{P}_0$ is a probability measure. Therefore, $f_\eta$ is dominated by an integrable envelope and the dominated convergence theorem applies. Since $f_\eta(z) \to 0$ pointwise as $\eta \to 0^+$, we obtain

$$\lim_{\eta \to 0^+} \mathbb{E}_{\mathbb{P}_0}[f_\eta(z)] = \mathbb{E}_{\mathbb{P}_0}\left[\lim_{\eta \to 0^+} f_\eta(z)\right] = \mathbb{E}_{\mathbb{P}_0}[0] = 0.$$

Since $\mathbb{E}_{\mathbb{P}_0}[f_\eta(z)] = \frac{1}{\eta}\mathbb{E}_{\mathbb{P}_0}[h(z)e^{-h(z)/\eta}]$, which proves (D). Combining (D) with (36) in (35) gives

$$\lim_{\eta \to 0^+} \left(\frac{\psi^{\min}}{\eta} - \frac{1}{\eta}\mathbb{E}_{\mathbb{Q}_\eta}[\psi]\right) = 0.$$

Finally, since

$$\lim_{\eta \to 0^+} \log \mathbb{E}_{\mathbb{P}_0}\left[e^{-(\psi(z) - \psi^{\min})/\eta}\right] = \log p^{\min},$$

we conclude that

$$\lim_{\eta \to 0^+} F'_\epsilon(\eta) = -\epsilon - \log p^{\min}. \tag{37}$$

Therefore, it follows from $\epsilon < -\log p^{\min}$ that $\lim_{\eta \to 0^+} F'_\epsilon(\eta) > 0$. On the other hand, we have $\lim_{\eta \to \infty} F'_\epsilon(\eta) < 0$. Furthermore, since $F''_\epsilon(\eta) < 0$ for all $\eta > 0$, the derivative $F'_\eta$ is strictly decreasing on $(\eta, \infty)$. Therefore, for $0 < \epsilon < -\log p^{\min}$, there exists a unique $\eta^*$ that $F'_\epsilon(\eta^*) = 0$ and $\eta^*$ is a global maximizer of $F_\epsilon(\eta)$ on $(0, \infty)$.

(iii) We have shown that for any $0 < \epsilon < -\log p^{\min}$, $F_\epsilon(\eta)$ admits a unique maximizer $\eta^* \in (0, \infty)$. We denote it by $\eta^*(\epsilon)$ when it is necessary to make this dependence explicit. To establish (25), we first show that for all $0 < \epsilon \leq \min\{-\frac{1}{2}\log p^{\min}, \epsilon_0\}$, the maximizer satisfies $\eta^*(\epsilon) \geq \eta_1$. Since $F'_\epsilon(\eta)$ is strictly decreasing in $\eta$ by (33), it suffices to verify that $F'_\epsilon(\eta_1) \geq 0$.

Since $\eta_1 \geq \eta_0 \geq M$, it follows from (21) and (22) that for all $\eta \geq \eta_1$,

$$F'_\epsilon(\eta) = \widetilde{F}'_\epsilon(\eta) + R'(\eta) = -\epsilon + \frac{\text{Var}_{\mathbb{P}_0}(\psi(z))}{2\eta^2} + R'(\eta). \tag{38}$$

Then by (23) and (24),

$$F'_\epsilon(\eta_1) \geq -\epsilon_0 + \frac{\text{Var}_{\mathbb{P}_0}(\psi(z))}{2(\eta_1)^2} - \frac{K_0}{(\eta_1)^3} \geq 0.$$

For $\widetilde{F}_\epsilon(\eta)$ defined in (22), let $\tilde{\eta}^*(\epsilon)$ denote its unique maximizer. We can solve for $\tilde{\eta}^*(\epsilon)$ in closed form by setting the first derivative to zero:

$$\widetilde{F}'(\eta; \epsilon) = -\epsilon + \frac{\text{Var}_{\mathbb{P}_0}(\psi)}{2\eta^2} = 0, \quad \text{i.e.} \quad \tilde{\eta}^*(\epsilon) = \sqrt{\frac{\text{Var}_{\mathbb{P}_0}(\psi)}{2\epsilon}}.$$

Since $F'_\epsilon(\eta^*) = \widetilde{F}'_\epsilon(\tilde{\eta}^*) = 0$, by (38) and (D), we obtain

$$\epsilon = \frac{\text{Var}_{\mathbb{P}_0}(\psi)}{2(\tilde{\eta}^*)^2} = \frac{\text{Var}_{\mathbb{P}_0}(\psi)}{2(\eta^*)^2} + R'(\eta^*).$$

Therefore,

$$\left(\frac{\eta^*}{\tilde{\eta}^*}\right)^2 = 1 + \frac{2R'(\eta^*)(\eta^*)^2}{\text{Var}_{\mathbb{P}_0}(\psi)}.$$

Since $\eta^*(\epsilon) \geq \eta_1$ for $0 < \epsilon \leq \epsilon_0$, it follows from (23) and (24) that

$$\left|\left(\frac{\eta^*}{\tilde{\eta}^*}\right)^2 - 1\right| = \frac{2|R'(\eta^*)|(\eta^*)^2}{\text{Var}_{\mathbb{P}_0}(\psi)} \leq \frac{2K_0}{\text{Var}_{\mathbb{P}_0}(\psi) \cdot \eta^*} \leq \frac{2K_0}{\text{Var}_{\mathbb{P}_0}(\psi) \cdot \eta_1} \leq \frac{1}{2}, \quad 0 < \epsilon \leq \epsilon_0,$$

which means

$$\sqrt{\frac{1}{2} \cdot \frac{\mathrm{Var}_{\mathbb{P}_0}(\psi)}{2\epsilon}} \le \eta^*(\epsilon) \le \sqrt{\frac{3}{2} \cdot \frac{\mathrm{Var}_{\mathbb{P}_0}(\psi)}{2\epsilon}}, \quad 0 < \epsilon \le \epsilon_0.$$

Then for $0 < \epsilon \le \epsilon_0$,

$$\sup_{\eta>0} F_\epsilon(\eta) = \widetilde{F}_\epsilon(\eta^*) + R(\eta^*) \le \widetilde{F}_\epsilon(\tilde{\eta}^*) + R(\eta^*)$$

$$\le \mathbb{E}_{\mathbb{P}_0}[\psi(z)] - \sqrt{2\epsilon \, \mathrm{Var}_{\mathbb{P}_0}(\psi)} + \frac{K_0}{(\eta^*)^2}$$

$$\le \mathbb{E}_{\mathbb{P}_0}[\psi(z)] - \sqrt{2\epsilon \, \mathrm{Var}_{\mathbb{P}_0}(\psi)} + 4K_0 \cdot \frac{\epsilon}{\mathrm{Var}_{\mathbb{P}_0}(\psi)}.$$

And

$$\sup_{\eta>0} F_\epsilon(\eta) \ge F_\epsilon(\tilde{\eta}^*) = \widetilde{F}_\epsilon(\tilde{\eta}^*) + R(\tilde{\eta}^*)$$

$$\ge \mathbb{E}_{\mathbb{P}_0}[\psi(z)] - \sqrt{2\epsilon \, \mathrm{Var}_{\mathbb{P}_0}(\psi)} - \frac{K_0}{(\tilde{\eta}^*)^2}$$

$$= \mathbb{E}_{\mathbb{P}_0}[\psi(z)] - \sqrt{2\epsilon \, \mathrm{Var}_{\mathbb{P}_0}(\psi)} - 2K_0 \cdot \frac{\epsilon}{\mathrm{Var}_{\mathbb{P}_0}(\psi)}.$$

Therefore, (25) holds when $0 < \epsilon \le \min\{-\frac{1}{2}\log p^{\min}, \epsilon_0\}$, which completes the proof of this theorem. $\qquad\square$

We now apply the general asymptotic result in Proposition D.2 to the KL formulation (18) derived in Theorem D.1. By identifying the robust surrogate $\psi_{\theta,\lambda}(z)$ with the random variable $\psi(z)$ and invoking the uniform boundedness conditions, we derive the following variance-regularized approximation as $\epsilon \to 0^+$.

**Corollary D.3.** *Consider the function $F_{\theta,\lambda}(\eta; \epsilon)$ given in the KL-Divergence dual problem* (18).

*Assume that there exists a constant $M > 0$ such that the moment generating function satisfies $\mathbb{E}_{\mathbb{P}_0}[e^{t\psi_{\theta,\lambda}(z)}] < \infty$ for all $t \in (0, 1/M]$ and $(\theta, \lambda) \in \Theta \times \mathbb{R}_+$. Consequently, the function admits the expansion*

$$F_{\theta,\lambda}(\eta; \epsilon) = \mathbb{E}_{\mathbb{P}_0}[\psi_{\theta,\lambda}(z)] - \eta\epsilon - \frac{\mathrm{Var}_{\mathbb{P}_0}(\psi_{\theta,\lambda}(z))}{2\eta} + R_{\theta,\lambda}(\eta).$$

*We assume that the remainder term $R_{\theta,\lambda}(\eta)$ satisfies uniform decay rates: there exist constants $K_0 > 0$ and $\eta_0 \ge M$ such that for all $\eta \ge \eta_0$ and $(\theta, \lambda) \in \Theta \times \mathbb{R}_+$,*

$$|R_{\theta,\lambda}(\eta)| \le \frac{K_0}{\eta^2}, \qquad |R'_{\theta,\lambda}(\eta)| \le \frac{K_0}{\eta^3}.$$

*Furthermore, let $\psi_{\theta,\lambda}^{\min} = \mathrm{ess\,inf}_{\mathbb{P}_0} \psi_{\theta,\lambda}(z)$ and $p_{\theta,\lambda}^{\min} = \mathbb{P}_0(\psi_{\theta,\lambda}(z) = \psi_{\theta,\lambda}^{\min})$. Assume that the variance and the probability mass are uniformly bounded from below:*

$$V_{\min} \triangleq \inf_{\theta,\lambda} \mathrm{Var}_{\mathbb{P}_0}(\psi_{\theta,\lambda}(z)) > 0 \quad \text{and} \quad p^{\min} \triangleq \inf_{\theta,\lambda} p_{\theta,\lambda}^{\min} > 0.$$

*Define the threshold parameters*

$$\epsilon_0 = \frac{V_{\min}}{2(\eta_1)^2} - \frac{K_0}{(\eta_1)^3}, \quad \text{with} \quad \eta_1 \triangleq \max\left\{\eta_0, \frac{4K_0}{V_{\min}}\right\}. \tag{39}$$

*Then for all $0 < \epsilon \le \min\{-\log p^{\min}, \epsilon_0\}$, the objective* (18) *admits the following simplified form:*

$$V_{\mathrm{KL}}(\mathbb{P}_0; \rho, \epsilon) = \inf_{\theta \in \Theta, \lambda \ge 0} \left\{ \lambda\rho + \mathbb{E}_{\mathbb{P}_0}[\psi_{\theta,\lambda}(z)] - \sqrt{2\epsilon \, \mathrm{Var}_{\mathbb{P}_0}(\psi_{\theta,\lambda}(z))} \right.$$

$$\left. + \mathcal{O}\left(\frac{\epsilon}{\mathrm{Var}_{\mathbb{P}_0}(\psi_{\theta,\lambda}(z))}\right) \right\}. \tag{40}$$

*Proof.* The proof proceeds by applying the general asymptotic analysis established in theorem D.2 to the specific parameterized family of functions involved in the exact KL-Divergence dual problem (18).

First, fix an arbitrary pair $(\theta, \lambda) \in \Theta \times \mathbb{R}_+$. The inner maximization problem in the exact dual (18) is given by:

$$\max \left\{ \sup_{\eta > 0} F_{\theta,\lambda}(\eta; \epsilon), \ \inf_{z \in \mathcal{Z}} \psi_{\theta,\lambda}(z) \right\}, \tag{41}$$

where $F_{\theta,\lambda}(\eta; \epsilon)$ is defined as in (20) with $\psi(z) = \psi_{\theta,\lambda}(z)$.

To simplify this structure, we analyze the behavior of $F_{\theta,\lambda}(\eta; \epsilon)$ as $\eta \to 0^+$. We derive upper and lower bounds for the logarithmic moment generating function term. For the lower bound, since $\psi_{\theta,\lambda}(z) \geq \psi_{\theta,\lambda}^{\min}$ almost surely, we have $\mathbb{E}_{\mathbb{P}_0}[e^{-\psi_{\theta,\lambda}(z)/\eta}] \leq e^{-\psi_{\theta,\lambda}^{\min}/\eta}$. Multiplying by $-\eta$ yields:

$$-\eta \log \mathbb{E}_{\mathbb{P}_0}[e^{-\psi_{\theta,\lambda}(z)/\eta}] \geq -\eta \left( -\frac{\psi_{\theta,\lambda}^{\min}}{\eta} \right) = \psi_{\theta,\lambda}^{\min}. \tag{42}$$

For the upper bound, by restricting the expectation to the set of minimizers and using the assumption $p_{\theta,\lambda}^{\min} > 0$, we obtain:

$$-\eta \log \mathbb{E}_{\mathbb{P}_0}[e^{-\psi_{\theta,\lambda}(z)/\eta}] \leq -\eta \log \left( p_{\theta,\lambda}^{\min} e^{-\frac{\psi_{\theta,\lambda}^{\min}}{\eta}} \right) = \psi_{\theta,\lambda}^{\min} - \eta \log p_{\theta,\lambda}^{\min}. \tag{43}$$

Combining (42) and (43), and noting that $-\eta\epsilon \to 0$ as $\eta \to 0^+$, we determine the limit:

$$\lim_{\eta \to 0^+} F_{\theta,\lambda}(\eta; \epsilon) = \psi_{\theta,\lambda}^{\min}. \tag{44}$$

Since the essential infimum dominates the absolute infimum (i.e., $\psi_{\theta,\lambda}^{\min} \geq \inf_{z \in \mathcal{Z}} \psi_{\theta,\lambda}(z)$), the limit (44) implies:

$$\sup_{\eta > 0} F_{\theta,\lambda}(\eta; \epsilon) \geq \lim_{\eta \to 0^+} F_{\theta,\lambda}(\eta; \epsilon) = \psi_{\theta,\lambda}^{\min} \geq \inf_{z \in \mathcal{Z}} \psi_{\theta,\lambda}(z). \tag{45}$$

Consequently, the first term in the maximization (41) is always greater than or equal to the second term, allowing us to reduce the problem equivalently to $\sup_{\eta > 0} F_{\theta,\lambda}(\eta; \epsilon)$. Thus, (18) can be written as

$$V_{\mathrm{KL}}(\mathbb{P}_0; \rho, \epsilon) = \inf_{\theta \in \Theta, \ \lambda \geq 0} \left\{ \lambda\rho + \sup_{\eta > 0} F_{\theta,\lambda}(\eta; \epsilon) \right\}, \tag{46}$$

It remains to show that the conditions required for the expansion (25) in theorem D.2(iii) are satisfied uniformly. The assumptions $V_{\min} > 0$ and $p^{\min} > 0$ guarantee that the conditions of theorem D.2(iii) are satisfied for all $(\theta, \lambda)$. Specifically, for any radius satisfying $0 < \epsilon \leq \min\{-\log p^{\min}, \epsilon_0\}$, the strict concavity and the second-order approximation error bounds hold uniformly. Therefore, for all $\theta, \lambda$, the expansion (25) holds for $F_{\theta,\lambda}(\eta; \epsilon)$. And substituting it into the simplified objective (46) completes the derivation of (40). $\qquad \square$

## E. Proof of Theorem 4.4

Under the given conditions, we know that Theorem 3.1 holds. The following analysis leverages the dual formulation (6) of $V_\phi(\mathbb{P}_0; \rho, \epsilon)$. The proof proceeds in two parts. We first establish the compactness of the optimal dual variables, and then bound the uniform convergence via Rademacher complexity.

First, fix $(\theta, \lambda)$ and consider the inner maximization problem of the dual objective (6) regarding $(\eta, \mu)$:

$$J(\eta, \mu) \triangleq \mu - \eta\epsilon - \eta\mathbb{E}_{\mathbb{P}_0} \left[ \phi^* \left( \frac{\mu - \psi_{\theta,\lambda}(z)}{\eta} \right) \right]. \tag{47}$$

We show that optimal solutions $(\eta^*, \mu^*)$ must be bounded.

For the analysis of the upper boundedness of $\eta$, we first establish a linear lower bound for the conjugate function $\phi^*$. By definition,

$$\phi^*(t) = \sup_{t \geq 0} \{ts - \phi(s)\}.$$

Evaluating the supremum at $t = 1$ and using the normalization $\phi(1) = 0$, we obtain $\phi^*(t) \geq t$, which yields

$$-\eta\phi^* \left( \frac{\mu - \psi_{\theta,\lambda}}{\eta} \right) \leq -\eta \left( \frac{\mu - \psi_{\theta,\lambda}}{\eta} \right) = -\mu + \psi_{\theta,\lambda}.$$

Substituting this into (47), we get $J(\eta, \mu) \leq \mathbb{E}_{\mathbb{P}_0}[\psi_{\theta,\lambda}(z)] - \eta\epsilon$. It follows from the definition (7) of $\psi_{\theta,\lambda}(z)$ that

$$\psi_{\theta,\lambda}(z) \leq \sup_{z' \in \mathcal{Z}} f_\theta(z').$$

Therefore, by Assumption 4.1, we have that $\psi_{\theta,\lambda}(z) \leq M$ for all $\theta \in \Theta, z \in Z$. Since $\epsilon > 0$ and $\mathbb{E}[\psi_{\theta,\lambda}] \leq M$, as $\eta \to \infty$, $J \to -\infty$. Thus, $\eta$ is upper bounded by some $\eta_{\max}$.

For the analysis of the boundedness of $\mu$, we examine the behavior of the dual objective $J(\eta, \mu)$ as $\mu$ diverges. As $\mu \to +\infty$, for any fixed $\eta > 0$ and bounded robust loss $\psi_{\theta,\lambda} \in [0, M]$, the argument $t = (\mu - \psi_{\theta,\lambda})/\eta$ tends to $+\infty$. By invoking Assumption 4.3, $\phi^*(t)$ is guaranteed to grow faster than any linear function. Specifically, for any slope $a > 1$, there exists a constant $b_a$ such that $\phi^*(t) \geq at - b_a$. By substituting this inequality into the dual objective (47), we observe that $J(\eta, \mu)$ is bounded above by a linear expression in $\mu$:

$$J(\eta, \mu) \leq \mu - \eta\epsilon - \eta\mathbb{E}_{\mathbb{P}_0} \left[ a \left( \frac{\mu - \psi_{\theta,\lambda}(z)}{\eta} \right) - b_a \right] = (1 - a)\mu + a\mathbb{E}_{\mathbb{P}_0}[\psi_{\theta,\lambda}(z)] + \eta(b_a - \epsilon).$$

Since $a > 1$, it follows that $J(\eta, \mu) \to -\infty$ as $\mu \to +\infty$.

Conversely, when $\mu \to -\infty$, the argument of the conjugate function $t = (\mu - \psi_{\theta,\lambda}(z))/\eta$ similarly diverges to $-\infty$ for any fixed $\eta > 0$. By definition, $\phi^*(t) = \sup_{s \geq 0}\{st - \phi(s)\}$. For standard divergences where $0$ is in the effective domain of $\phi$, we have the constant lower bound $\phi^*(t) \geq 0 \cdot t - \phi(0) = -\phi(0)$. This reveals that while $t$ decreases, the expectation term $\eta\mathbb{E}_{\mathbb{P}_0}[\phi^*(t)]$ in the objective remains bounded from below by the constant $-\eta\phi(0)$. Consequently, the dual objective satisfies the asymptotic upper bound:

$$J(\eta, \mu) = \mu - \eta\epsilon - \eta\mathbb{E}_{\mathbb{P}_0} \left[ \phi^* \left( \frac{\mu - \psi_{\theta,\lambda}(z)}{\eta} \right) \right] \leq \mu - \eta(\epsilon - \phi(0)).$$

As $\mu \to -\infty$, the linear term $\mu$ dominates the expression, driving $J(\eta, \mu)$ to $-\infty$. This two-sided divergence ensures that the optimal dual variable $\mu^*$ is contained within a compact set, satisfying the necessary conditions for establishing the generalization certificate via Rademacher complexity.

For the analysis of the lower boundedness of $\eta$, we establish that the dual objective $J(\eta, \mu)$ diverges to $-\infty$ as $\eta \to 0^+$, ensuring that the optimal $\eta^*$ is bounded away from zero. We define the random variable $Y_z = (\psi_{\theta,\lambda}(z) - \mu)^2$. We invoke Assumption 4.1, which guarantees that the robust loss has a uniformly non-degenerate variance, i.e., $\mathrm{Var}_{\mathbb{P}_0}(\psi_{\theta,\lambda}(z)) \geq \sigma^2 > 0$ for all $\theta \in \Theta$ and $\lambda \in [0, M/\rho]$. This implies that for any fixed $\mu$, the second moment satisfies $\mathbb{E}_{\mathbb{P}_0}[Y_z] \geq \mathrm{Var}_{\mathbb{P}_0}(\psi_{\theta,\lambda}(z)) \geq \sigma^2$. Applying the Paley-Zygmund inequality, there exist uniform constants $\delta_0 > 0$ and $p_0 > 0$ such that the probability of the robust loss deviating from $\mu$ is bounded from below:

$$\mathbb{P}_0 \left( \mu - \psi_{\theta,\lambda}(z) \geq \delta_0 \right) \geq p_0.$$

Specifically, the variance ensures the distribution supports values smaller than $\mu$, so we consider the event $E = \{z : \mu - \psi_{\theta,\lambda}(z) \geq \delta_0\}$ with $\mathbb{P}_0(E) \geq p_0$.

Consider $\eta$ sufficiently small such that for any $z \in E$, the argument of the conjugate function satisfies $t = \frac{\mu - \psi_{\theta,\lambda}(z)}{\eta} \geq \frac{\delta_0}{\eta} \geq t_0$. Under this condition, we apply Assumption 4.3 which states that $\phi^*(t) \geq c_1 t^{1+q} - c_2$ for $t > t_0$. We can thus upper-bound the expectation term in the dual objective as

$$-\eta \, \mathbb{E}_{\mathbb{P}_0} \left[ \phi^* \left( \frac{\mu - \psi_{\theta,\lambda}(z)}{\eta} \right) \right] \leq -\eta \int_E \left( c_1 \left( \frac{\delta_0}{\eta} \right)^{1+q} - c_2 \right) d\mathbb{P}_0(z)$$

$$\leq -\frac{c_1 \delta_0^{1+q} p_0}{\eta^q} + \eta \, c_2 \, p_0.$$

Since $q > 0$, as $\eta \to 0^+$, the leading term proportional to $-1/\eta^q$ diverges to $-\infty$. This forces the dual objective $J(\eta, \mu) \to -\infty$, implying that the maximum cannot be achieved arbitrarily close to zero. Consequently, there exists a uniform lower bound $\eta_{\min} > 0$ such that the optimal dual variable satisfies $\eta^* \geq \eta_{\min}$.

Finally, we establish the boundedness of $\lambda$. Note that the primal objective function value is equivalent to (11). By the definition (7) of the robust loss, we have $\psi_{\theta,\lambda}(z) \geq f_\theta(z) - \lambda c(z, z) = f_\theta(z)$. Since $f_\theta \geq 0$, it follows that $\psi_{\theta,\lambda}(z) \geq 0$ for all $z$, which implies that the inner distributional risk $\inf_{\mathbb{P}_D} \mathbb{E}_{\mathbb{P}_D}[\psi_{\theta,\lambda}]$ is non-negative. Consequently, the total objective is lower bounded by $\lambda\rho$. Given $\rho > 0$, as $\lambda \to \infty$, the objective value tends to $+\infty$. However, choosing a feasible $\lambda = 0$ yields a finite objective value (bounded by $M$). Therefore, to minimize the objective, the optimal $\lambda^*$ must reside within a compact interval $[0, \Lambda_{\max}]$.

Having established that the optimal dual variables $(\lambda^*, \eta^*, \mu^*)$ lie in a compact set, denoted as $\mathcal{K}$, we restrict the optimization domain to $\Xi_{\mathcal{K}} \triangleq \Theta \times \mathcal{K}$. Since $\Theta$ is compact, it follows that $\Xi_{\mathcal{K}}$ is also a compact set. We then proceed to bound the generalization error.

Let the full dual objective function in (6) with respect to a distribution $\mathbb{P}$ be defined as:

$$Q(\xi; \mathbb{P}) \triangleq \underbrace{\lambda\rho + \mu - \eta\epsilon}_{C(\xi)} + \mathbb{E}_{z \sim \mathbb{P}}[g_\xi(z)],$$

where $g_\xi(z) \triangleq -\eta\phi^* \left( \frac{\mu - \psi_{\theta,\lambda}(z)}{\eta} \right)$ and $\xi = (\theta, \lambda, \eta, \mu) \in \Xi_{\mathcal{K}}$. Note that the term $C(\xi)$ depends only on the parameters and is independent of the data distribution.

We aim to bound the difference in optimal values:

$$\Delta \triangleq \left| V_\phi(\mathbb{P}_0; \rho, \epsilon) - V_\phi(\hat{\mathbb{P}}_n; \rho, \epsilon) \right| = \left| \inf_{\theta,\lambda} \sup_{\eta,\mu} Q(\xi; \mathbb{P}_0) - \inf_{\theta,\lambda} \sup_{\eta,\mu} Q(\xi; \hat{\mathbb{P}}_n) \right|.$$

Using the elementary inequality $|\sup_x A(x) - \sup_x B(x)| \leq \sup_x |A(x) - B(x)|$, which analogously holds for the infimum operator, we can sequentially move the supremum operator outside the absolute value:

$$\begin{aligned}
\Delta &\leq \sup_{\theta,\lambda} \left| \sup_{\eta,\mu} Q(\xi; \mathbb{P}_0) - \sup_{\eta,\mu} Q(\xi; \hat{\mathbb{P}}_n) \right| \\
&\leq \sup_{\theta,\lambda} \sup_{\eta,\mu} \left| Q(\xi; \mathbb{P}_0) - Q(\xi; \hat{\mathbb{P}}_n) \right| \\
&= \sup_{\xi \in \Xi_{\mathcal{K}}} \left| (C(\xi) + \mathbb{E}_{\mathbb{P}_0}[g_\xi]) - (C(\xi) + \mathbb{E}_{\hat{\mathbb{P}}_n}[g_\xi]) \right|.
\end{aligned}$$

Observe that the deterministic term $C(\xi)$ cancels out, reducing the problem to bounding the uniform deviation of the function class $\mathcal{G} \triangleq \{g_\xi \mid \xi \in \Xi_{\mathcal{K}}\}$:

$$\left| V_\phi(\mathbb{P}_0; \rho, \epsilon) - V_\phi(\hat{\mathbb{P}}_n; \rho, \epsilon) \right| \leq \sup_{\xi \in \Xi_{\mathcal{K}}} \left| \mathbb{E}_{\mathbb{P}_0}[g_\xi(z)] - \mathbb{E}_{\hat{\mathbb{P}}_n}[g_\xi(z)] \right|. \tag{48}$$

By standard learning theory results, for a sample size $n$ and any confidence parameter $\delta \in (0, 1)$, the uniform deviation is bounded with probability at least $1 - \delta$ by the empirical Rademacher complexity:

$$\sup_{\xi \in \Xi_{\mathcal{K}}} \left| \mathbb{E}_{\mathbb{P}_0}[g_\xi(z)] - \mathbb{E}_{\hat{\mathbb{P}}_n}[g_\xi(z)] \right| \leq 2\mathfrak{R}_n(\mathcal{G}) + M_{\mathcal{G}} \sqrt{\frac{\log(2/\delta)}{2n}}, \tag{49}$$

where $M_{\mathcal{G}}$ is the uniform bound of the function class $\mathcal{G}$, whose existence is guaranteed by the compactness of $\mathcal{K}$ and Assumption 4.1.

To analyze $\mathfrak{R}_n(\mathcal{G})$, we verify the Lipschitz property to apply Talagrand's Contraction Lemma. For any fixed dual parameters $(\eta, \mu)$, define the scalar function $h_{\eta,\mu} : \mathbb{R} \to \mathbb{R}$ as:

$$h_{\eta,\mu}(u) \triangleq -\eta\phi^* \left( \frac{\mu - u}{\eta} \right).$$

The function class can be viewed as the composition $\mathcal{G} = \{z \mapsto h_{\eta,\mu}(\psi_{\theta,\lambda}(z))\}$. We compute the derivative of $h_{\eta,\mu}(\cdot)$ with respect to its input $u$:

$$\left| \frac{d}{du} h_{\eta,\mu}(u) \right| = \left| -\eta \cdot (\phi^*)' \left( \frac{\mu - u}{\eta} \right) \cdot \left( -\frac{1}{\eta} \right) \right| = \left| (\phi^*)' \left( \frac{\mu - u}{\eta} \right) \right|.$$

Since $(\eta, \mu)$ are restricted to a compact set and $\psi_{\theta,\lambda}$ is bounded, the argument $t = \frac{\mu - \psi_{\theta,\lambda}}{\eta}$ lies within a bounded domain $\mathcal{D} \subset \mathbb{R}$. By Assumption 4.2, $\phi^*$ is $L_{\phi^*}$-Lipschitz on compact sets, which implies its derivative is bounded: $|(\phi^*)'(t)| \leq L_{\phi^*}$ for all $t \in \mathcal{D}$. Thus, for any $(\eta, \mu) \in \mathcal{K}$, the function $h_{\eta,\mu}(\cdot)$ is $L_{\phi^*}$-Lipschitz. Applying Talagrand's Contraction Lemma yields:

$$\mathfrak{R}_n(\mathcal{G}) \leq \sup_{(\eta,\mu) \in \mathcal{K}} L_{h_{\eta,\mu}} \mathfrak{R}_n(\Psi_{\Theta,\Lambda}) = L_{\phi^*} \mathfrak{R}_n(\Psi_{\Theta,\Lambda}).$$

Substituting this complexity bound back into (49) completes the proof.

## F. Experimental Details

We implemented all methods using PyTorch 2.0.0 on Python 3.9. All experiments were conducted with one NVIDIA GeForce RTX 3090 GPU. **Synthetic Experiments.** We optimized the model parameters using SGD with a learning rate of $\eta_\theta = 0.01$ and a dual learning rate of $\eta_\lambda = 0.01$. The inner maximization was solved via $K = 5$ steps of PGD with a step size of $\eta_{pgd} = 0.1$. Since the dataset size is moderate, we utilized full-batch training to ensure stable gradient estimation. All OT-based methods utilize the squared Euclidean distance as the ground cost. To ensure a fair comparison of the underlying mechanisms, we fixed the radius of the inner maximization constraint at $\rho = 1.0$ for all methods. All experiments were repeated over 10 independent trials with different random seeds.

Specifically, for the sample efficiency experiments with fixed dimension $d = 5$ shown in Figure 1b, we employed a fixed divergence radius $\epsilon = 0.1$ for nested DRO. We calibrated method specific hyperparameters for the baselines to optimize their respective performance. We set the robustness radius $\epsilon = 0.4$ for OR-WDRO and set the regularization parameters to $\lambda = 10.0$ and $\beta = 10.0$ for UOT-DRO.

For the high dimensional robustness experiments shown in Figure 1c, where $d$ ranges from 10 to 50, we calibrated the hyperparameters for both the OR-WDRO and UOT-DRO baselines via a grid search on representative dimensions of 10 and 50. The optimal configurations obtained at these two dimensions were then averaged and used as a unified set of hyperparameters across all experimental dimensions. Consequently, we set the robustness radius $\epsilon = 0.4$ for OR-WDRO and set the regularization parameters to $\lambda = 20.0$ and $\beta = 10.0$ for UOT-DRO in these high dimensional experiments. Specifically, for nested DRO, the magnitude of loss statistics fluctuates significantly in high-dimensional regimes, rendering a fixed $\epsilon$ suboptimal. Leveraging the variance regularization property established in (10), we introduced an adaptive radius strategy governed by a safety ratio $r_t$. This metric quantifies the relative strength of the cleaning gain with respect to the expected loss at training step $t$:

$$r_t \triangleq \frac{\sqrt{2\epsilon_t \cdot \mathrm{Var}(\psi_t)}}{\mathbb{E}[\psi_t]},$$

where $\psi_t$ denotes the robust surrogate loss in the current batch. Instead of utilizing a static $\epsilon$, we employed a dynamic calibration strategy to maintain the ratio between the variance-driven cleaning gain and the mean loss at a fixed target level. Specifically, we calibrated the radius to achieve a target safety ratio of $r_{target} = 0.4$ in this experiment. At each training step $t$, the radius $\epsilon_t$ is computed as

$$\epsilon_t = \frac{(r_{target} \cdot \mathbb{E}_{\mathbb{P}_n}[\psi])^2}{2 \cdot \mathrm{Var}_{\mathbb{P}_n}(\psi) + \delta},$$

where $\delta = 10^{-8}$ is a numerical stability constant. This mechanism ensures the variance-driven cleaning gain consistently accounts for 40% of the mean loss. As shown in Figure 1c, this adaptive mechanism allows nested DRO to maintain stability as dimensions increase.

**CIFAR-10 Experiments.** We optimized the model parameters using SGD with an initial learning rate of $\eta_\theta = 0.05$, a momentum coefficient of 0.9, and a weight decay of 5e-4. The learning rate was decayed using a cosine annealing scheduler over 80 training epochs with a batch size of $B = 128$. For all CIFAR-10 results reported in the paper, we report best clean, defined as the highest accuracy attained on the clean CIFAR-10 test set over the course of training.

For the geometric layer, we model transformations on $\mathcal{G} = SO(2)$ and define the transport cost using the rotation angle relative to the identity. Concretely, for a transformed sample $z' = (T_\alpha(x), y)$, we use

$$c(z', z) = \left( \frac{|\alpha|}{\theta_{\max}} \right)^2, \qquad \theta_{\max} = 90°,$$

which corresponds to a normalized squared geodesic cost on $SO(2)$ with labels fixed throughout the transformation. The robust surrogate $\psi_{\theta,\lambda}(z)$ is then computed by optimizing over a continuous rotation angle. The geometric radius is fixed at $\rho = 1.0$, and the dual variable $\lambda$ is optimized jointly with the network parameters. For nested DRO, the adaptive outer-radius strategy from the main text uses target ratio $r_{\text{target}} = 0.12$, minimum radius $10^{-3}$, and maximum radius $1.0$.

For the CIFAR-10 corruption protocol, we partition the training set into four disjoint groups: 10% valid geometric samples, 20% overlap-corrupted samples, 10% plain label-noise samples, and the remaining 60% clean samples. For each sample in the valid-geometric and overlap groups, we draw a single rotation angle whose magnitude is sampled uniformly from $[60°, 120°]$ and whose sign is chosen uniformly at random, and keep that angle fixed throughout training. Overlap-corrupted samples additionally receive a random incorrect label, whereas plain label-noise samples receive a random incorrect label without any rotation.

The comparison methods are implemented as follows. Statistics-only applies trimming directly to per-sample cross-entropy losses. Sequential robustify-then-trim first computes the geometry-robust surrogate and then trims those surrogate values. Nested DRO optimizes the full nested objective. We additionally report adapted versions of OR-WDRO and UOT-DRO on CIFAR-10. Since these methods were developed primarily for lower-dimensional settings and do not come with standard deep-image training pipelines, they were not part of our initial CIFAR-10 comparison. To provide a more complete empirical comparison, we add them here using high-dimensional adaptations that preserve their core robust-optimization structures rather than literal reproductions of the original algorithms. Our OR-WDRO adaptation operates on minibatch penultimate-layer representations together with per-sample losses and optimizes the same regularized CVaR-type robust objective, while our UOT-DRO adaptation preserves its exponential-aggregation structure in learned feature space. In the implementation used for Table 2, we set the OR-WDRO robustness parameter to $\epsilon = 0.30$ and the UOT-DRO hyperparameters to $(\lambda, \beta, \lambda_2) = (10, 10, 10)$.

