# OpenReview forum: "Robust Learning via Nested Distributionally Robust Optimization"
_ICML.cc/2026/Conference — ICML 2026 regular_

### Official Review · Reviewer_44zs · 2026-02-25

**Soundness:** 3
**Presentation:** 3
**Significance:** 3
**Originality:** 2
**Overall Recommendation:** 4
**Confidence:** 2

**Summary:**

Proposes a Nested Distributionally Robust Optimization (Nested DRO) framework to resolve the "geometric conflation" problem when feature perturbations and statistical contamination co-exist.

**Compliance With Llm Reviewing Policy:**

Affirmed.

**Final Justification:**

The authors attempted to address my concerns. I am maintaining my score.

**Key Questions For Authors:**

The theoretical contributions are outstanding and rigorously derived, but there are major gaps in the comparative experiments on real-world data.

**Limitations:**

Yes

**Strengths And Weaknesses:**

Achieves strict decoupling of compound uncertainty via an inner Optimal Transport layer and an outer $f$-divergence layer.

Successfully derives a computable strong dual formulation for this bilevel, non-convex problem. Establishes a rigorous generalization bound using Rademacher complexity. Rigorously proves that the framework (under $\chi^2$-divergence) is equivalent to variance-regularized risk minimization, providing an excellent theoretical foundation for outlier suppression.

Missing Baseline Comparisons: The real-world CIFAR-10 experiment only compares against ERM, completely omitting the state-of-the-art baselines evaluated in the synthetic experiments.

The empirical evaluation on real-world scenarios is restricted to a single dataset (CIFAR-10), making the evidence somewhat thin.

---

> ### Author Rebuttal · Authors · 2026-03-31
>
> **Response to the Main Empirical Weakness**
>
> We thank the reviewer for the positive assessment of our theoretical contributions and for pointing out the main empirical weakness of the current submission. We agree that the original real-data evaluation was too limited: in the submitted version, the CIFAR-10 study only compared against ERM, whereas the synthetic experiments included geometry-coupled DRO baselines such as OR-WDRO and UOT-DRO. We also agree that relying on a single real-data benchmark makes the empirical evidence thinner than ideal.
>
> In the original submission, we did not include OR-WDRO and UOT-DRO on CIFAR-10 because these methods were developed primarily for lower-dimensional settings and do not come with standard deep-image training pipelines. We therefore wanted to avoid presenting an ad hoc implementation that might not faithfully reflect their intended robust-optimization mechanisms. To address this concern more carefully in the rebuttal, we implemented high-dimensional adaptations that preserve the core structure of both methods, so that the comparison better reflects their intended robust-optimization structure.
>
> To strengthen the rebuttal accordingly, we substantially expanded the CIFAR-10 study. First, we replaced the original discrete rotation search with a stronger continuous geometric version using differentiable rotations and PGD-based inner maximization over continuous $SO(2)$, which better matches the continuous-cost formulation in Section 3. Second, we added **CIFAR-10 comparisons against OR-WDRO and UOT-DRO using the high-dimensional adaptations described above. We also added a Statistics-only baseline**, which performs loss-based filtering without any geometric robustification, in order to isolate the contribution beyond pure sample rejection.
>
> In the new corrupted CIFAR-10 setting with 10% geometric perturbation, 10% label-flip contamination, and 20% overlap contamination, we report the highest clean-test accuracy attained during training under the same evaluation protocol for all methods. Under this metric, **Nested DRO achieves 78.78%**, compared with 75.00% for ERM, 70.11% for OR-WDRO, 74.10% for UOT-DRO, and 71.50% for Statistics-only. These results show that the benefit of Nested DRO is not explained by pure loss-based filtering alone, while also providing a much more comprehensive real-data comparison than in the original submission.
>
> Regarding the breadth of real-data evaluation, we agree that one dataset is still a limitation. Within the rebuttal timeframe, we prioritized substantially strengthening CIFAR-10 rather than adding another dataset with limited tuning, since CIFAR-10 is our main real-data testbed for mixed geometric and statistical corruption. In the revised paper, we will incorporate the expanded continuous-$SO(2)$ benchmark, the added OR-WDRO/UOT-DRO and Statistics-only comparisons, and the single-shift ablations, and we will also moderate the empirical claims to reflect that the strongest current real-data evidence is on CIFAR-10. Overall, we hope these additions address the reviewer’s concern that the theory is strong while the real-data comparison was previously incomplete.

---

> > ### Author Rebuttal · Reviewer_44zs · 2026-04-05
> >
> > I thank authors to their responses and their commitment of revision.  Some issues could not be resolved during the rebuttal phase. I will maintain my score.

---

> > > ### Author Response · Authors · 2026-04-08
> > >
> > > Thank you very much for your careful consideration of our rebuttal. We fully respect your judgment and sincerely appreciate your time and thoughtful feedback!

---

### Official Review · Reviewer_F5Bc · 2026-03-09

**Soundness:** 3
**Presentation:** 4
**Significance:** 4
**Originality:** 3
**Overall Recommendation:** 4
**Confidence:** 4

**Summary:**

Distributionally robust optimization (DRO) is used to improve the model when the data distribution may shift from the original one. Here, the paper focuses on the fact that the two representative distributional shifts require different formulation of DRO: geometric perturbations and statistical contaminations. If these two shifts co-exist but we apply a DRO method considering only either, it leads to performance degradation. The paper first formulates the problem as "nested" DRO, thus tri-level optimization (one for model parameter optimization, and the other two for distributional uncertainty). Then the paper shows (1) the problem can be converted into a tractable bi-level optimization, and (2) derived the convergence guarantee when solving the problem with an empirical form (replacing the distribution with the summation for all samples).

**Compliance With Llm Reviewing Policy:**

Affirmed.

**Key Questions For Authors:**

- Section 1.2: Although the conceptual advantage of the proposed method compared to existing methods employing Wasserstein distance and criteria to suppress the effect of statistical outliers is discussed in this section, what are the computational and/or algorithmic novelty? Perhaps employing the bi-level distributional optimization (tri-level including the model parameter optimization) is novel?
- Section 5:
  - What if the proposed method is applied to a dataset having either geometric or statistical distribution change? It would be the best if the proposed method works well for such a case, and is examined experimentally.
  - As far as reading Appendix F, $\\rho$ and $\\epsilon$ are assumed to be predetermined. However, I think that they should not be fixed since they are data-dependent. In this sense, the reviewer suggest that the either of the following examinations should be done: (1) estimating $\\rho$ and $\\epsilon$ (and similar hyperparameters in other methods) from data, or (2) varying $\\rho$ and $\\epsilon$ at the optimal ones and around, and measureing the effect of the misspecification of $\\rho$ and $\\epsilon$.
  - In Section 5.2, why the comparison is made only between NDRO and ERM? Are the methods in Section 5.1 (standard DRO, OR-WDRO and UOT-DRO) unavailable for this experiment?

---- **The followings are minor questions**

- Overall: In this problem formulation (Section 3.1), the optimization is an ordinary DRO with respect to $\\mathbb{P}\_W$, but not an ordinary DRO with respect to $\\mathbb{P}\_D$ (this can rather be called as constrained optimization), so calling both as DRO is somewhat confusing. Can there be better naming for this bi-level problem?
- Section 5.2: Is "ERM" stands for "Empirical Risk Minimization" (i.e., no treatment of robustness)? If so, please write so.
- Appendix C: If $\\phi(t)=\\frac{1}{2}(t - 1)^2$, its Fenchel conjugate is just $\\psi^*(s) = \\frac{1}{2}s^2 + s$, not the one specified in the manuscript consisting of two cases. How the Fenchel conjugate in the manuscript is derived? (Perhaps an additional constraint is requried?)

**Limitations:**

As stated in "Key Questions For Authors" part, although experimental performance when the hyperparameters of the proposed method is set up to the true distributional shift is examined, the performance when the hyperparameters is estimated is not examined. So the reviewer thinks that additional examinations are needed to be applied to real-life uses.

No negative societal impact looks arise.

**Strengths And Weaknesses:**

Strengths:

- [Originality][Significance] It is known that, thanks to Kantorovich-Rubinstein duality (Appendix A), a function optimization with respect to a data distribution under Wasserstein distance constraint (which is intractable) can be converted into an optimization with a penalty of expectation of distances for the data distribution (like did in (Li et al., 2025)). This paper showed that, even if we add a constraint to represent statistical contaminations as another level of optimization in $\\phi$-divergence, a tractable computation can be derived.
- [Soundness][Originality] Algorithmically, we can optimize the model by replacing the expectation for the data distribution above with the summation for all samples or minibatch samples. This makes the optimization easy, and a theoretical upper bound of the target optimization function is derived.

Weaknesses:

- [Significance] The reviewer felt that the discussions of (dis)advantages of the proposed method compared to similar existing methods are unclear. Please see "Section 1.2" part in "Key Questions For Authors" section.
- [Soundness][Significance] The reviewer felt that the experimental evaluations are insufficient. Please see "Section 5" part in "Key Questions For Authors" section.

---

> ### Author Rebuttal · Authors · 2026-03-31
>
> **Response to Weakness 1 and Key Question 1**
>
> We thank the reviewer for this thoughtful question. The main novelty of our work is the decoupled nested formulation together with its tractable dual instantiation. Unlike OR-WDRO and UOT-DRO, which couple geometric perturbation handling and outlier suppression in a single ambiguity set, we formulate the problem as a bilevel distributional optimization, with an inner OT layer for geometric robustness and an outer $\phi$-divergence layer for statistical cleaning. To the best of our knowledge, this decoupled bilevel robust model is new.
>
> The key computational contribution is that this nested formulation is not only conceptual: through dualization, it yields a practical training objective that can be optimized with standard minibatch procedures, and in the $\chi^2$ case it further induces an explicit variance-based cleaning term. This gives a concrete optimization target that separates geometric smoothing from loss-based reweighting, rather than relying on a single coupled robustification mechanism. We will revise Section 1.2 to make this distinction more explicit.
>
> **Response to Key Question 2.1**
>
> We thank the reviewer for this suggestion. We added single-shift ablations on CIFAR-10 to evaluate the method when only one type of distribution change is present. Under 40% geometric corruption without statistical contamination, **Nested DRO achieves 89.01%**, compared with 83.03% for ERM, 79.95% for OR-WDRO, and 83.97% for UOT-DRO, showing that the method remains strong even when the shift is purely geometric. Under 40% label noise without geometric corruption, Nested DRO still improves over ERM (71.47% vs. 68.45%), although the margin is smaller. These results suggest that the proposed decoupling is most beneficial when geometric and statistical effects co-exist, while in purely statistical settings the gain is more limited. We will revise the paper to make this scope clearer and avoid overstating the method’s advantage outside the compound-uncertainty regime.
>
> **Response to Key Question 2.2**
>
> Thank you for this suggestion. We followed the reviewer’s second suggestion and added a local misspecification analysis for $(\rho, \epsilon)$ on the synthetic benchmark. Concretely, we first identified a reference setting from a coarse sweep and then ran a denser local sweep around that point to evaluate sensitivity to nearby misspecification.
>
> The results suggest a reasonably broad stable region rather than a sharply tuned optimum. Near the selected region, performance changes only mildly with $\rho$. We also observe that smaller-than-selected $\epsilon$ values remain stable in a substantial neighborhood, although they are somewhat less effective. By contrast, the main sensitivity appears when $\epsilon$ is pushed too large: at $\epsilon$=0.25, we begin to observe unstable or outlier runs, and the degradation becomes much more severe by $\epsilon$=0.30. Overall, these results suggest that the method is fairly robust near the chosen hyperparameters, while overly aggressive $\epsilon$ can move training outside the stable regime. We will clarify this local sensitivity analysis in the revision.
>
> **Response to Key Question 2.3**
>
> We thank the reviewer for this question. In the original submission, Section 5.2 compared Nested DRO only against ERM because OR-WDRO and UOT-DRO were originally developed for lower-dimensional settings and do not come with standard deep-image training pipelines. To provide a more complete comparison, we now add CIFAR-10 results for both methods using high-dimensional adaptations that preserve their core robust-optimization structures. In the corrupted CIFAR setting, **Nested DRO achieves 78.78%**, outperforming OR-WDRO (70.11%), UOT-DRO (74.10%), and ERM (75.00%). These additional results strengthen the empirical comparison in Section 5.2.  For more details, see response to Reviewer jTmG (Weaknesses 2&3 and Key Question 1).
>
> **Response to Key Questions 3&4&5**
>
> We thank the reviewer for these helpful comments.
>
> (1) We agree that the terminology in the current draft may be imprecise. The inner layer is a standard worst-case OT-DRO step, whereas the outer layer is a statistical cleaning step rather than another worst-case DRO layer. We will revise Section 3.1 to make this distinction clearer and consider a more accurate term for the overall formulation.
>
> (2) Yes, ERM stands for Empirical Risk Minimization. We will spell this out at its first appearance in Section 5.2/Table 2.
>
> (3) The conjugate in our paper is defined as $\phi^*(s) = \sup_{t \ge 0} \{st - \phi(t)\}$, since $t = \frac{dP}{dP_0}$ is a density ratio and must therefore be nonnegative. Under this domain restriction, for $\phi(t) = \frac{1}{2}(t-1)^2$, the conjugate is the piecewise form given in the paper, rather than $\frac{1}{2}s^2 + s$ over all of $\mathbb{R}$. No extra primal constraint is needed beyond nonnegativity of the density ratio. We will revise to make this point more explicit.

---

> > ### Author Rebuttal · Reviewer_F5Bc · 2026-04-04
> >
> > Thank you for responses, including additional experimental examinations. My concerns became clear.

---

> > > ### Author Response · Authors · 2026-04-04
> > >
> > > Thank you very much for carefully considering our additional results and clarifications. We fully respect your judgment and sincerely appreciate your feedback, which helped us improve the paper. If you believe the rebuttal has strengthened the work beyond your current assessment, we would be grateful if you would consider reflecting that in your final score. Thank you again for your time and consideration.

---

### Official Review · Reviewer_hDvZ · 2026-03-12

**Soundness:** 2
**Presentation:** 3
**Significance:** 2
**Originality:** 2
**Overall Recommendation:** 4
**Confidence:** 3

**Summary:**

This paper proposes a nested DRO framework combining an inner optimal transport (OT) ambiguity set for geometric robustness with an outer $\phi$-divergence constraint for statistical cleaning. The authors derive a tractable dual, establish equivalence to variance-regularized risk minimization, provide Rademacher-based generalization bounds, and present experiments on synthetic regression and CIFAR-10.

**Compliance With Llm Reviewing Policy:**

Affirmed.

**Final Justification:**

Concern addressed.

**Key Questions For Authors:**

- How does the nested framework perform compared to standard adversarial training followed by loss-based trimming (e.g., DORO or simple top-k removal)?
- Can you construct a setting where loss-based filtering alone fails but the nested formulation succeeds — thereby demonstrating that the joint optimization is essential?
- Can you explain how the empirical formula for $r$ and $\epsilon$ in Appendix F? It would be helpful in understanding the calibration of the method.

**Limitations:**

yes

**Strengths And Weaknesses:**

**Strengths**
- Well-articulated problem. The "geometric conflation" phenomenon — where OT-based methods cannot distinguish valid high-leverage points from outliers — is a real issue, and the paper frames it clearly. The synthetic visualization in Figure 1a is effective.
- Clear presentation of the main method. The nested construction is natural, well-motivated by the "geometric conflation" phenomenon. The method part is well-written and easy to follow.
- Nice insight on the method. Theorem 3.2 gives a closed form of the intermediate distribution, which helps understand the reweighting function of the outer optimization. Theorem 3.3 provides nice insight by connecting outlier suppression and variance regularization.

**Weekness**
- My biggest concern is with the effectiveness of the method. From the experiment, I suspect the core working mechanism is loss-based sample rejection, and directly setting the Wasserstein radius to 0 would give you similar results. The authors should compare with the methods on loss-based sample rejection, like setting Wasserstein radius to 0 in the proposed method, [1] as mentioned in the Related Works, and classic methods like [2] and [3].
- The "Optimistic Reweighting" can discard genuinely hard clean samples. The paper criticizes geometry-based methods for conflating valid high-leverage points with outliers, but the loss-based reweighting in the outer layer introduces an analogous conflation: genuinely difficult clean samples with high robust loss may be suppressed alongside true contaminants. The paper assumes that high robust loss implies statistical anomaly, but this is precisely the kind of single-criterion conflation the paper criticizes in other approaches — just applied in loss space rather than feature space. No experiment tests this failure mode.
- The computation issue. The algorithm requires K PGD steps per sample per iteration. No wall-clock comparisons are provided. The CIFAR-10 experiment sidesteps continuous PGD entirely by using exhaustive search over four discrete rotations — a simplification that may not extend to realistic geometric perturbation models.

[1] Zhai, Runtian, Chen Dan, Zico Kolter, and Pradeep Ravikumar. "Doro: Distributional and outlier robust optimization." In International Conference on Machine Learning, pp. 12345-12355. PMLR, 2021.

[2] Li, Tian, Ahmad Beirami, Maziar Sanjabi, and Virginia Smith. "Tilted empirical risk minimization." arXiv preprint arXiv:2007.01162 (2020).

[3] Han, Bo, Quanming Yao, Xingrui Yu, Gang Niu, Miao Xu, Weihua Hu, Ivor Tsang, and Masashi Sugiyama. "Co-teaching: Robust training of deep neural networks with extremely noisy labels." Advances in neural information processing systems 31 (2018).

---

> ### Author Rebuttal · Authors · 2026-03-31
>
> **Response to Weakness 1 and Key Questions 1&2**
>
> We thank the reviewer for this important concern. To directly test whether the proposed method is primarily acting as a loss-based sample rejection mechanism, we added a new rebuttal experiment on CIFAR-10 under an overlap-confounded
> **continuous $SO(2)$** setting. The training data contains three corruption types: 10% valid geometric perturbations only (rotated but correctly labeled samples), 10% label corruption only (label-flipped but unrotated samples), and 20% overlap corruption (simultaneously rotated and label-flipped samples).
>
> We also make the baselines explicit. Statistics-only performs only loss-based filtering on the original inputs, without any geometric robustification. Sequential two-stage first applies continuous $SO(2)$ adversarial training and then performs trimming on the resulting robust-surrogate scores. Thus, these baselines correspond to pure filtering and a robustify-then-filter pipeline, respectively.
>
> We report the highest clean-test accuracy attained during training for all methods under the same evaluation protocol. Under this metric, **Nested DRO achieves 78.78%**, outperforming Statistics-only (71.50%), Sequential two-stage (74.65%) and ERM (75.00%). This gap relative to Statistics-only isolates the benefit beyond pure loss filtering, while the gap relative to Sequential two-stage shows that the gain is not explained by a simple robustify-then-filter pipeline. Together, these results support the claim that the joint nested formulation itself provides a clear additional benefit.
>
> **Response to Weakness 2**
>
> We thank the reviewer for this insightful comment. We agree that the outer layer of NDRO is not a perfect anomaly detector. It downweights samples with high robust surrogate loss, and therefore some intrinsically difficult but clean samples may also be downweighted when high robust loss does not perfectly align with statistical contamination.
>
> That said, our claim is narrower. Nested DRO is designed to decouple statistical validity from geometry, rather than to perfectly separate every hard clean sample from every contaminant. Our synthetic experiment was designed to test precisely this geometry-conflation regime, and the results support this more specific claim. We will revise the paper to make this scope explicit and avoid presenting the method as a universal anomaly detector.
>
> **Response to Key Question 3**
>
> Thank you for this question. In Appendix F, $r_t$ is introduced only as a calibration variable, not as an additional modeling assumption. Its role is to normalize the cleaning strength relative to the current robust-loss scale. Concretely, rather than fixing $\epsilon$ directly, we fix a target ratio $r_{target}$ for the variance-based cleaning term relative to the current mean robust loss, and then recompute $\epsilon_t$ from the current batch mean and variance so that this relative cleaning strength remains approximately stable during training.
>
> The motivation is practical rather than theoretical: a fixed $\epsilon$ can become too weak or too aggressive as the scale of $\psi_t$ changes across iterations or across experimental regimes. The adaptive rule keeps the cleaning effect on a comparable scale throughout training and improves optimization stability. We will revise Appendix F to state this motivation more explicitly.
>
> **Response to Weakness 3**
>
> We thank the reviewer for highlighting the computational issue. We agree that methods with an inner robust optimization step are more expensive than standard ERM, and that this overhead should be reported explicitly. We also agree that the earlier CIFAR-10 implementation, which evaluated only four discrete rotations, was a simplification of the continuous geometric formulation.
>
> To address this directly in the rebuttal, we replaced that implementation with a continuous $SO(2)$ version using differentiable image rotations and PGD-based inner maximization over the rotation angle.  In particular, Standard DRO, Sequential two-stage, and Nested DRO all use the same continuous $SO(2)$ inner search.
>
> We further added a wall-clock comparison on CIFAR-10. Measured on a single NVIDIA GeForce RTX 3090 GPU (PyTorch 2.4.0, CUDA 12.4; Intel Xeon Silver 4210 CPU), using batch size 128, we report the average training epoch time over 3 timed epochs after 1 warm-up epoch. The average epoch time is **19.45s for ERM**, 110.66s for Standard DRO, 106.47s for Sequential two-stage, and **106.08s for Nested DRO**. Hence, the continuous robust-training variants are approximately  $5.5\times$ slower than ERM in our setup.
>
> While the overhead is substantial, it improves theory–implementation fidelity: the updated procedure now instantiates the continuous formulation directly, rather than approximating it by a discrete four-rotation surrogate. We will include this wall-clock comparison in the revision and clarify that scaling to richer perturbation families is an important direction for future work.

---

> > ### Author Rebuttal · Reviewer_hDvZ · 2026-04-03
> >
> > Thanks for the authors' response. I appreciate the authors' effort in comparing with the statistics-only method. I wonder what the statistics-only method used in this experiment is? Is the Nested DRO fixing the Wasserstein radius to 0?
> >
> > My second concern is not addressed; we still don't know when the Nested DRO brings more benefit over filtering out clean high-loss samples. The authors claim the Nested DRO method is targeted at the geometry-conflation regime. But what kinds of data distributions and decision-making tasks are in this regime is not immediately clear. The two experiments in the paper are not informative enough for me to see the common feature of the regime where Nested DRO works well.

---

> > > ### Author Response · Authors · 2026-04-03
> > >
> > > We thank the reviewer for the follow-up questions. We realize that this point was not stated clearly enough in our rebuttal, and we clarify it below.
> > >
> > > ## Response to your question
> > >
> > > The *statistics-only* baseline in our rebuttal is in fact a **pure loss-filtering method**. Concretely, it computes per-sample cross-entropy on the original inputs and trims the highest-loss fraction before the SGD update, i.e., it uses a simple trimmed-loss filtering rule. In contrast, the Nested DRO results in that comparison use the full geometric layer, with radius set to 1.0.
> > >
> > > ## Response to your concern
> > > It is helpful to distinguish loss-filtering methods from DRO-based methods. **Plain loss-filtering** methods identify suspicious samples purely through their training losses on the original inputs, and discard or downweight the highest-loss portion. As a result, they do not distinguish whether a sample has high loss because it is genuinely corrupted or because it is a clean but difficult example arising from valid geometric variation.
> > >
> > > **Standard DRO**, by contrast, is well suited to settings where the dominant uncertainty comes from localized or adversarial input perturbations, i.e., valid geometric variation around observed samples. This is the perspective emphasized in [SND18], which casts adversarial robustness through a distributionally robust optimization lens.
> > >
> > > Subsequent variants extend this geometric-DRO viewpoint to contaminated data. In particular, **OR-WDRO** [NGS23] is motivated by the observation that standard WDRO captures geometric uncertainty such as sampling noise or localized perturbations, but is less well suited to non-geometric contamination such as adversarial outliers; it therefore augments WDRO to handle both Wasserstein perturbations and contamination. **UOT-DRO** [WSZ24] similarly proposes an ambiguity model inspired by unbalanced optimal transport, with a soft penalization mechanism designed to be more resilient to outliers.
> > >
> > > **Our motivation is that**, in settings with geometric perturbations, harmful contaminated samples are not necessarily geometrically outlying, while clean samples may still be difficult because of valid geometric transformations. Consequently, these two sources of difficulty are not reliably separable by plain loss alone. Moreover, they cannot always be adequately addressed by a single geometry-coupled mechanism, nor by a simple sequential combination of geometric robustification and filtering. Our goal is precisely to address this setting, where valid geometric hardness and statistical contamination are tightly entangled despite playing fundamentally different roles. This is why we introduce a nested two-layer formulation.
> > >
> > > We will revise the paper to make this scope more explicit, and we hope the above clarification resolves your concern. We fully respect your judgment, and we hope that this clarification will help you more fully assess the contribution of our method in the regime it is designed to address. We sincerely thank you again for prompting us to make this point more precise.
> > >
> > > ---
> > >
> > > [SND18] Sinha, A., Namkoong, H., and Duchi, J. (2018). *Certifying Some Distributional Robustness with Principled Adversarial Training*. International Conference on Learning Representations.
> > >
> > > [NGS23] Nietert, S., Goldfeld, Z., and Shafiee, S. (2023). *Outlier-Robust Wasserstein DRO*. Advances in Neural Information Processing Systems, 36, 62792–62820.
> > >
> > > [WSZ24] Wang, Z., Shen, Y., Zavlanos, M. M., et al. (2024). *Outlier-Robust Distributionally Robust Optimization via Unbalanced Optimal Transport*. Advances in Neural Information Processing Systems, 37, 52189–52214.

---

### Official Review · Reviewer_jTmG · 2026-03-12

**Soundness:** 3
**Presentation:** 3
**Significance:** 2
**Originality:** 3
**Overall Recommendation:** 4
**Confidence:** 4

**Summary:**

This paper proposed a nested DRO problem (eq.4), including three layers: one $\theta$-learning layer, one $P_D$-learning layer subject to $P_D$ being nearby nominal $P_0$, and one $P_W$-learning layer subject to $P_W$ being nearby $P_D$. This minimax problem is reformulated as a (bilevel constrained) finite-dimensional problem (eq.6) using duality. In particular, when $\phi$ is the $\chi^2$ divergence, the outer $P_D$-learning admits a closed-form optimizer (thm3.2(i)) and eq.4 reduces to eq.10. The variance regularizer appears with a negative sign as authors considered "the best" $P_D$ , in contrast with "the worst" $P_W$.  Algorithm 1 deploys and iteratively updates adversarial samples via PGD (7+8), outer objective (13) and minimizes $(\theta,\lambda)$ via GD (14+15).

**Compliance With Llm Reviewing Policy:**

Affirmed.

**Final Justification:**

The authors addressed my concerns and committed to revise accordingly. It changed my evaluation from 3: Weak reject to 4: Weak accept.

**Key Questions For Authors:**

1.The weaknesses on continuity assumption of $c$ and comprehensive benchmarking experiments are critical to re-assess the contribution of this work.

2. The cost function $c$ (eq.5) "prohibits transport across different labels". However in the whole section 5, "20% of the samples are corrupted with symmetric label noise". Is this mismatch intentional?

3. Do boundedness and non-degenerate variance (assumption 4.1.) hold for the linear hypothesis in section 5.1?

4. Is there a connection between the idea of nested DRO and the bilevel DRO proposed in [3]?

[3] Per-Group Distributionally Robust Optimization (Per-GDRO) with Learnable Ambiguity Set Sizes via Bilevel Optimization, S Jung, W Lee, J Hamm, J Park, OPT 2025: Optimization for Machine Learning

**Limitations:**

The authors do not discuss any potential negative societal impact of their work.

**Strengths And Weaknesses:**

1. **Strengths.** The theoretical claims are technically sound and correct. The theoretical parts of the paper are well-written and easy to follow, with context and prior literature are mostly discussed. The work provides new insights on how to decouple geometric smoothing from statistical cleaning in DRO learning. In particular, it shows how the closed-form (distribution) solution of the stastiscal learning problem can benefit and calibrate the worst-case distribution in robust learning. It also reveals that despite of stacking one more optimization level, the resulting problem has the same generalization bound as the standard one via Rademacher complexity.

2. **Weakness.**
- It is unclear why decoupling into optimistic statistical learning and pessimistic geometric is *realistic*. On one hand, this work involves *minimizing* statistical cleaning. By strict complementary condition, $\epsilon$  is small and the optimal $P_D$ is exactly $\epsilon$-away from $P_0$. On the other hand, existing approaches use *positive* variance regularization, thus intuitively involving *maximizing* statistical learning.
- The tractable dual problem (eq.6) requires $c$to be continuous. However, $c$ is discrete(w.r.t. the rotation) in the CIFAR-10 task (sec.5.1.).
- The proposed NDRO surpassed standard DRO, OR-WDRO and UOT-DRO on the synthetic task (sec.5.1.), however it was compared to ERM on ResNet-18 only in the CIFAR-10 task (sec.5.2). In order to assess the efficacy of NDRO, it is crucial to also benchmark it against the above methods and other standard adversarial training methods such as FSGM [2] or AutoAttack [1], and with architectures  other than only ResNet-18.

[1] Reliable evaluation of adversarial robustness with an ensemble of diverse parameter-free attacks, F Croce, M Hein, ICML 2020
[2] Explaining and harnessing adversarial examples, IJ Goodfellow, J Shlens, C Szegedy, ICLR 2015

---

> ### Author Rebuttal · Authors · 2026-03-31
>
> **Response to Weakness 1**
>
> We thank the reviewer for raising this concern. Our outer layer is not intended as a worst-case statistical robustness layer; rather, it is a contamination-cleaning layer. The modeling motivation is that statistically anomalous samples, such as mislabeled examples, should not be absorbed by the geometric layer. If all such effects are handled only through geometric DRO, the method may misinterpret statistical contamination as geometric perturbation and overly penalize informative hard examples. In our formulation, the inner OT layer handles local geometric perturbations, while the outer $\phi$-divergence layer performs statistical cleaning through reweighting. **This is why the two optimization directions differ**: the inner layer takes a supremum because geometric perturbations are modeled adversarially, whereas the outer layer takes an infimum because it selects a cleaner intermediate distribution rather than a worst-case statistical adversary. Accordingly, the induced variance term appears with a negative sign: it reflects optimistic cleaning of contaminated samples, not worst-case statistical regularization. We will revise the paper to make this modeling intent more explicit.
>
> **Response to Weaknesses 2&3 and Key Question 1**
>
> We thank the reviewer for highlighting the continuity and benchmarking issues. The reviewer is correct that the submitted CIFAR experiment used a discrete search, whereas the theory in Sec. 3 is stated for a continuous cost function. More precisely, Appendix F defines the geometric model on the transformation manifold $SO(2)$, while the original CIFAR implementation used an exhaustive search over a finite cyclic subgroup only as a simplification. To remove this concern, in the rebuttal we replace that implementation with a stronger CIFAR-10 version using differentiable rotations and PGD-based inner maximization **over continuous $SO(2)$ rotations**, directly matching the continuous-cost formulation in Sec. 3. We also agree that the original CIFAR benchmark, which only compared against ERM, was not sufficiently comprehensive. Since OR-WDRO and UOT-DRO were developed mainly for lower-dimensional settings and do not come with standard deep-image training pipelines, we did not include them in the original submission. To provide a more complete empirical comparison, we now add CIFAR-10 results against both methods using faithful high-dimensional adaptations that preserve their core robust-optimization structures, rather than a literal reproduction of the original algorithms. In the new corrupted CIFAR setting (10% geometric perturbation, 10% label-flip contamination, and 20% overlap contamination), we report the highest clean-test accuracy reached during training. Under this metric,
> **Nested DRO attains 78.78%**, outperforming OR-WDRO (70.11%), UOT-DRO (74.10%), and ERM (75.00%). These additions directly address the reviewer’s main concern by aligning the CIFAR inner maximization with the continuous-cost theory and by making the empirical benchmarking substantially more comprehensive.
>
> **Response to Key Question 2**
>
> We thank the reviewer for pointing out this subtle issue. In Eq. (5), the ground cost $c(z,z') = c_{\mathcal{X}}(x, x') + \infty \cdot \mathbf{1}_{\{y \neq y'\}}$ constrains the inner OT coupling to preserve the observed label, so geometric transport is only allowed within a label class. By contrast, the “symmetric label noise” in the CIFAR-10 experiment refers to statistical corruption in the observed training labels. This type of contamination is handled by the outer statistical-cleaning layer, rather than by allowing cross-label transport in the inner OT problem. We will revise the paper to make this distinction clearer.
>
> **Response to Key Question 3**
>
> The reviewer is correct that Assumption 4.1 is stronger than what is explicitly enforced in Sec. 5.1. Assumption 4.1 requires uniformly bounded loss and uniformly non-degenerate variance, whereas the synthetic linear experiment in Sec. 5.1 is presented as an empirical stress test and does not explicitly impose truncation or projection to guarantee these conditions. We will revise the paper to clarify this distinction: Sec. 4 provides a generalization certificate under regularity assumptions, while Sec. 5.1 is an empirical evaluation not intended as a direct verification of those assumptions.
>
> **Response to Key Question 4**
>
> We thank the reviewer for pointing us to [3]. Our method was developed independently, and we agree that [3] is conceptually related to our work, since both consider hierarchical combinations of robustness mechanisms. Our formulation focuses on sample-level decoupling between statistical cleaning and geometric smoothing, whereas [3] studies group-structured robustness with bilevel learning of ambiguity sizes. We will add this citation and clarify the connection in the revised version.

---

> > ### Author Rebuttal · Reviewer_jTmG · 2026-04-04
> >
> > I thank authors to their responses and their commitment of revision. I will raise score accordingly.

---

> > > ### Author Response · Authors · 2026-04-04
> > >
> > > Thank you so much for your encouraging reply! We are absolutely delighted to hear that our rebuttal has successfully addressed your concerns. Your positive feedback truly means a lot to us, and we greatly appreciate the time and effort you’ve dedicated to reviewing our work.

---

### Decision · Program_Chairs · 2026-04-30

**Decision:**

Accept (regular)

**Comment:**

The paper proposes a novel variant of Distributionally Robust Optimization (DRO) that incorporates an outer "optimistic" step. Specifically, in addition to the standard supremum (worst-case) taken over a Wasserstein-based neighborhood of distributions, the authors introduce an outer infimum (best-case) over a neighborhood defined by f-divergence. The intuition behind this formulation is that the outer approximation serves as a mechanism for statistical cleaning, allowing the model to mitigate the impact of corrupted data before the inner robustness step is applied.

The reviewers agree that the formulation is novel and that the technical contributions regarding tractable reformulations are mathematically strong.

However, there were shared concerns regarding the empirical validation and the conceptual interpretation of the proposed framework. The clarifications provided during the rebuttal are helpful and should be incorporated into the final version of the paper. Given that the motivation for the optimistic step is statistical cleaning, the paper should discuss  connection to the robust statistics literature at a deeper level (see, for example, [1]) and to Rockafellian relaxations. Concretely, it would be interesting to discuss what happens when the $\phi$-divergence is instantiated as total variation distance, a canonical model in robust statistics. Finally, what happens when one interchanges the infimum and supremum order? Does it lead to anything meaningful?